# IFN-β is a macrophage-derived effector cytokine facilitating the resolution of bacterial inflammation

Senthil Kumaran Satyanarayanan [1,5], Driss El Kebir[2,5], Soaad Soboh[1,5], Sergei Butenko [1], Meriem Sekheri[2], Janan Saadi[1], Neta Peled [1], Simaan Assi[1], Amira Othman[2], Sagie Schif-Zuck [1], Yonatan Feuermann[3], Dalit Barkan[1], Noa Sher[4], János G. Filep [2] & Amiram Ariel[1]

The uptake of apoptotic polymorphonuclear cells (PMN) by macrophages is critical for timely resolution of inflammation. High-burden uptake of apoptotic cells is associated with loss of phagocytosis in resolution phase macrophages. Here, using a transcriptomic analysis of macrophage subsets, we show that non-phagocytic resolution phase macrophages express a distinct IFN-β-related gene signature in mice. We also report elevated levels of IFN-β in peritoneal and broncho-alveolar exudates in mice during the resolution of peritonitis and pneumonia, respectively. Elimination of endogenous IFN-β impairs, whereas treatment with exogenous IFN-β enhances, bacterial clearance, PMN apoptosis, efferocytosis and macro-phage reprogramming. STAT3 signalling in response to IFN-β promotes apoptosis of human PMNs. Finally, uptake of apoptotic cells promotes loss of phagocytic capacity in macrophages alongside decreased surface expression of efferocytic receptors in vivo. Collectively, these results identify IFN-β produced by resolution phase macrophages as an effector cytokine in resolving bacterial inflammation.

[1] Department of Biology and Human Biology, University of Haifa, Haifa 3498838, Israel. [2] Department of Pathology and Cell Biology, University of Montreal, and Research Center, Maisonneuve-Rosemont Hospital, Montreal, QC H1T 2M4, Canada. [3] ResCure Pharma, Haifa 3498838, Israel. [4] Tauber Bioinformatics Center, University of Haifa, Haifa 3498838, Israel. [5]These authors contributed equally: Senthil Kumaran Satyanarayanan, Driss El Kebir, Soaad Soboh. Correspondence and requests for materials should be addressed to J.G.F. (email: janos.g.filep@umontreal.ca) or to A.A. (email: amiram@research.haifa.ac.il)

Timely resolution of acute inflammation involves apoptotic death of tissue infiltrating neutrophils and their removal by neighboring macrophages to restore organ structure and function[1–3]. Macrophages recognize apoptotic cells through unique 'eat me' signals and cognate receptors and engulf them in an efficient and tightly regulated-manner to avoid autoantigen exposure and metabolic stress[4,5]. In turn, the efferocytic macrophages undergo reprogramming that diverts proinflammatory cells to cells that drive tissue repair and resolution[6–8]. A unique subset of non-phagocytic pro-resolving macrophages is generated during resolving inflammation following high-burden efferocytosis and exposure to pro-resolving mediators[9–11]. These CD11b[low] macrophages exhibit increased expression of 12/15-lipoxygenase (LO), an enzyme involved in the synthesis of specialized pro-resolving lipid mediators (SPMs)[12], reduced arginase 1 expression and emigration to distant sites[9].

Neutrophil apoptosis is perceived as one of the control points that limits the number of neutrophils at sites of inflammation and pushes ongoing inflammation towards resolution. Inflammatory cytokines, growth factors and bacterial constituents, such as GM-CSF, LPS and bacterial DNA (CpG DNA) can prolong the lifespan of neutrophils by delaying apoptosis[13–15] predominantly by preserving the expression of the BCL2 family protein Mcl-1. By contrast, pro-resolving mediators, such as 15-epi-LXA₄, annexin A1, resolvin E1, and cyclin-dependent kinase inhibitors[1,16–19] facilitate degradation of Mcl-1 and consequently promote neutrophil apoptosis.

The role of type I interferons in host response to viruses is well established, whereas the contribution of these cytokines to antibacterial defense is less clear. IFN-β has been reported to limit immune responses in some experimental models of inflammation through induction of IL-10 and/or inhibition of the inflammasome and IL-1β production[20–24]. IFN-β was also found to block cancer progression by limiting the recruitment of pro-angiogenic neutrophils into tumors and deletion of endogenous IFN-β was associated with delayed apoptosis of tumor-associated neutrophils[25,26]. By contrast, IFN-β accelerates monocytic inflammation by attracting Ly-6C⁺ monocytes to sites of chronic inflammation[27]. Enhanced IFN-β expression was detected in DNAse II-deficient embryonic macrophages that lack phagolysosomal degradation of DNA and hence accumulate large amounts of DNA from apoptotic cells[28], thereby limiting erythrocyte differentiation and leading to severe anemia[28]. Thus, IFN-β may exert important bioactivities consistent with a role in resolving inflammation.

Here we use an unbiased transcriptomic analysis to identify IFN-β as an effector cytokine that is produced and acting during resolution of peritonitis and *E.coli*-evoked pneumonia. Mechanistically, IFN-β overrides pro-survival cues and promotes apoptosis in murine and human neutrophils through STAT3 activation, and enhances efferocytosis by resolution phase macrophages, leading to the reprogramming of these macrophages to an anti-inflammatory and pro-resolving phenotype. Finally, we show that non-phagocytic macrophages are generated exclusively following the uptake of apoptotic cells in an irreversible manner, and this conversion is associated with reduced surface expression of efferocytic receptors. These findings identify IFN-β as a macrophage-derived multi-pronged effector in resolving inflammation.

## Results

### Non-phagocytic macrophages have a distinct transcriptome.
Since non-phagocytic macrophages compose a significant portion of resolution phase macrophages, we aimed to identify the molecular characteristics that distinguish these cells from their phagocytic counterparts. To this end, we performed transcriptome analysis of PKH2[hi] (phagocytic) and PKH2[lo/−] (non-phagocytic) F4/80⁺ macrophages sorted from resolving exudates (Fig. 1a).

Comparative analysis of these two macrophage populations revealed disparate regulation of 3045 genes (≥2-fold change among 13368 genes expressed in either subset). Of these genes, 1511 were upregulated and 1534 were downregulated in non-phagocytic macrophages. We performed gene ontology annotation to characterize the functional groups of the genes affected (Fig. 1b). As expected, non-phagocytic macrophages downregulated phagocytosis-associated functions, such as endocytosis, membrane invagination, small GTPase signaling and cell adhesion. Notably, non-phagocytic macrophages displayed significant metabolic changes, including increases in oxidative phosphorylation and oxidation-reduction processes. Non-phagocytic macrophages also showed reductions in gene groups associated with tissue repair, such as blood vessel development and morphogenesis, extracellular matrix, protein kinase activity and cell adhesion (required for wound retraction), while increasing locomotor activity. These findings are consistent with the origin of non-phagocytic CD11b[low] macrophages that differentiate from M2–like macrophages during the resolution of inflammation[9] and with their capacity to migrate to distant sites. These macrophage subsets expressed low levels of markers for peritoneal resident macrophages, such as GATA6 (1.37 and 0.19 RPKM for phagocytic and non-phagocytic macrophages, respectively), Tim4 (relative values of 3.47 and 1.26 RPKM, respectively), and TGF-β2 (relative values of 0.25 and 0.01 RPKM, respectively) with no detectable levels of another marker, GPR37[29,30]. Moreover, F4/80⁺Tim4⁺ macrophages were essentially absent at 66 h PPI in WT mice, while they were the major resident macrophage population in unchallenged mice (Supplementary Fig. 1), as previously reported[31]. Both subsets did express high levels of the monocyte markers CD115 (424.49 and 217.42 RPKM, respectively), and CD74 (1733.47 and 2631.85 RPKM, respectively), indicating that they originated exclusively from monocyte-derived macrophages rather than yolk sack-originated resident macrophages.

### Non-phagocytic macrophages express IFN-β-associated genes.
Our transcriptome analysis also revealed that non-phagocytic macrophages express a distinct protein signature profile (Table 1) consistent with upregulated IFN-β expression and activity[32]. This profile is similar to that observed following macrophage activation due to impaired DNA degradation. These genes included *Ifnb1, Ifitm1, Il1b, Isg15, Cxcl9, Ifit1, Ifit3, Ccl2, Isg20, Ifitm6, Ccl3, Ccl5* and *Cxcl10*. To confirm IFN-β-associated activation of non-phagocytic macrophages, we compared the expression of IFN-β and its target gene *Isg15* in non-phagocytic and phagocytic macrophages. Western blotting of peritoneal macrophages and cell-free exudates from *Ifnb*⁻/⁻ mice revealed the existence of two different species of IFN-β (Supplementary Fig. 2a, b), a 50–66 kDa isoform that is the most prevalent in resident and resolution phase macrophages, and a 25–35 kDa isoform that prevails in the exudates, which was scarcely detectable in macrophages. Notably, all the secreted isoforms were diminished at 16 h PPI upon clodronate-mediated depletion of resident peritoneal macrophages (Supplementary Fig. 2c). However, peritoneal IFN-β levels were not abrogated by administering clodronate-containing liposomes during the resolution phase (Supplementary Fig. 2c–d), probably due to IFN-β secretion by non-phagocytic macrophages that cannot be depleted by clodronate. Furthermore, during the resolution phase, we detected intracellular IFN-β protein only in F4/80⁺ macrophages and not in lymphocytes or

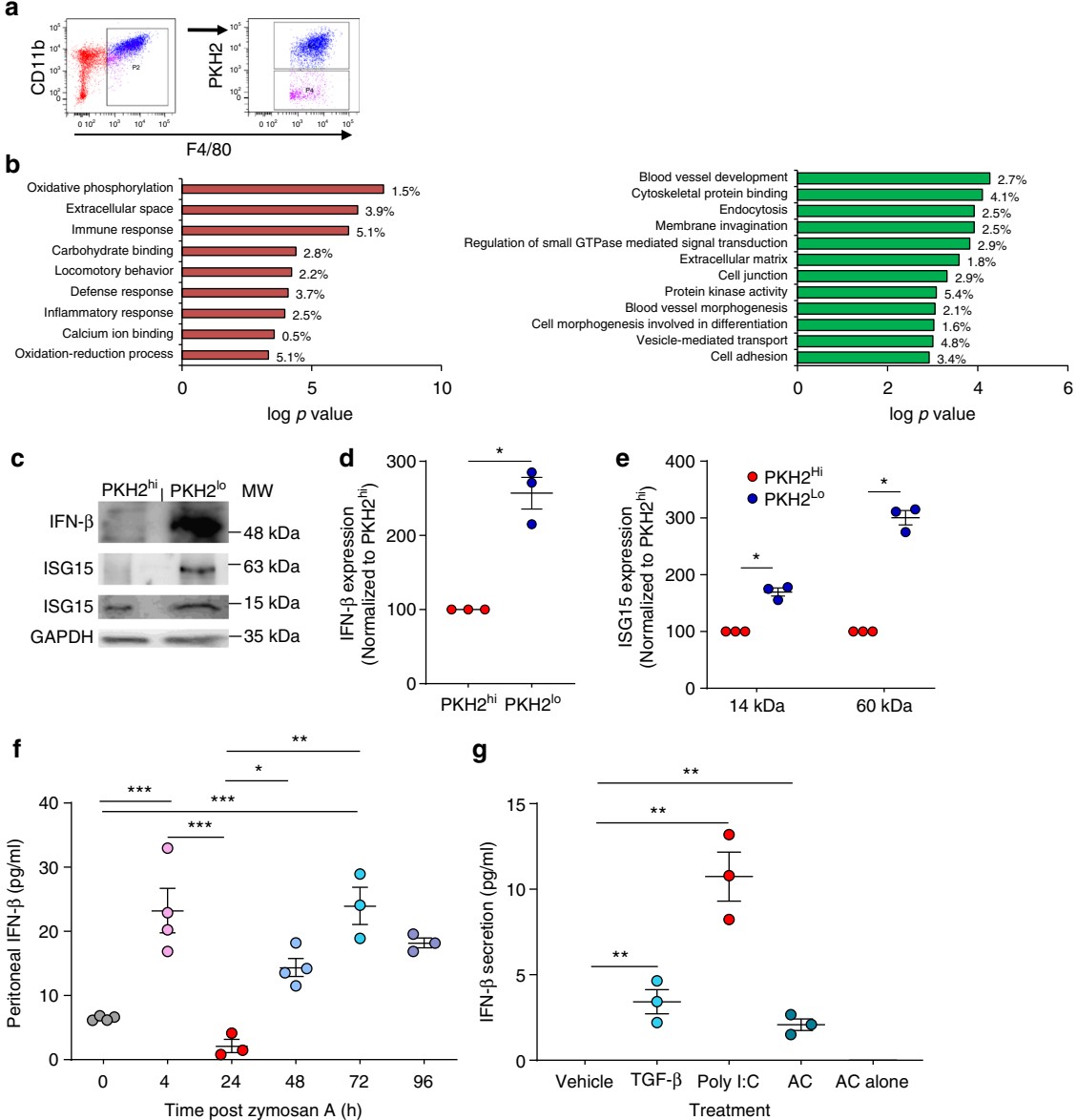

**Fig. 1** Non-phagocytic resolution phase macrophages express IFN-β. **a**, **b** Male mice were injected intraperitoneally with zymosan A (1 mg/mouse) followed by an injection of PKH2-PCL at 62 h. After 4 h, the peritoneal cells were recovered and immuno-stained for F4/80 and CD11b. Then, F4/80+ macrophages were sorted based on the extent of PKH2-PCL acquisition (PKH2 + /− populations; >98% purity) using the FACSAria III sorter (illustrated in (**a**). The collected cells were immediately used for RNA extraction (with RNA integrity value above 7.5), and a gene expression microarray analysis was performed using Illumina hiSeq 2500. Differential gene expression analysis and gene ontology (GO) enrichment were performed for genes that were significantly upregulated (**b**, left panel) or downregulated (**b**, right panel) in non-phagocytic/satiated (PKH2-PCL^lo) macrophages in comparison to phagocytic (PKH2-PCL^hi) ones. The results indicate the statistical significance of the GO term and the percentage of enrichment is presented. **c–e** Expression of IFN-β and ISG15 in sorted satiated and phagocytic macrophages. Representative results (**c**) and mean ± SEM (**d**, **e**) for three independent experiments. *P < 0.05 (Student's t test). **g**, **h** Peritoneal exudates were collected from unchallenged mice (0 h) or following peritonitis for 4–96 h. IFN-β content in cell-free fluids was determined by ELISA (**f**). Results are mean ± SEM from three (24, 72, 96 h) or four (0, 4, 48 h) mice. *P < 0.05, **P < 0.01, ***P < 0.005 (Tukey's HSD). Alternatively, resolution phase macrophages were recovered 66 h post peritonitis initiation (PPI) and incubated with TGF-β (5 ng/ml), poly (I:C) (4 μg/ml) or apoptotic cells (AC, at a ratio of 1:5) for 24 h. Culture supernatants were then collected and IFN-β content was measured (**g**). Culture media from apoptotic cells served as control. Results are representative from three independent experiments. *P < 0.05, **P < 0.01, ***P < 0.005 (Tukey's HSD). Source data are provided as a Source Data file

eosinophils (Supplementary Fig. 2e). *Ifnb*^−/− macrophages had the same background staining as the 2^nd antibody control. Moreover, the 25–35 kDa isoforms and, to a higher extent, the 50 kDa isoform were present in both resident and resolution phase macrophages, whereas the 66 kDa isoform was exclusively expressed in resolution phase macrophages (Supplementary Fig. 2f). Analysis of organelle fractions of RAW264.7

macrophages that were incubated with apoptotic cells revealed presence of the 50 and 66 kDa isoforms of IFN-β in the ER and secretory vesicles (which contain calnexin and rab7, respectively) (Supplementary Fig. 2g). The 25–35 kDa isoforms are present in the ER and negligible in secretory vesicles. These results suggest that the 25 kDa IFN-β would undergo posttranslational modifications in the ER that would prevent its secretion. Once these

**Table 1 Non-phagocytic macrophages exhibit an IFN-β-related gene signature**

| Gene | PKH2$^{hi}$ | | PKH2$^{lo}$ | | Fold change (log2) | Q value |
|------|-------------|------|-------------|------|--------------------|---------|
| Ifnb1 | −0.34 | 0.33 | 3.26 | 4.38 | 3.89 | 0.0011 |
| Ifitm1 | 6.21 | 7.00 | 8.67 | 9.47 | 2.47 | 0.0024 |
| Il1b | 11.20 | 10.99 | 13.67 | 13.34 | 2.41 | 0.0006 |
| Isg15 | 7.12 | 6.07 | 9.09 | 8.23 | 2.03 | 0.0006 |
| Cxcl9 | 2.19 | 1.35 | 3.99 | 3.60 | 1.97 | 0.0006 |
| Ifit1 | 5.06 | 3.15 | 6.82 | 5.12 | 1.80 | 0.0006 |
| Ifit3 | 1.45 | 0.34 | 3.31 | 1.98 | 1.79 | 0.0006 |
| Ccl2 | 8.55 | 8.75 | 10.04 | 10.50 | 1.64 | 0.0159 |
| Isg20 | 3.12 | 3.13 | 4.48 | 4.99 | 1.63 | 0.0006 |
| Ifitm6 | 7.69 | 7.81 | 8.88 | 9.70 | 1.60 | 0.0062 |
| Ccl3 | 8.85 | 8.77 | 9.86 | 10.30 | 1.28 | 0.0006 |
| Ccl5 | 8.37 | 7.16 | 9.55 | 8.13 | 1.12 | 0.0006 |
| Cxcl10 | 7.60 | 7.58 | 8.08 | 8.89 | 0.95 | 0.0006 |

Log2 of RPKM values

modifications are removed, IFN-β is immediately secreted, hence the 25–35 kDa isoforms are the only forms detected consistently in peritoneal fluids, but scarcely inside macrophages. Notably, non-phagocytic macrophages (PKH2$^{lo}$) expressed significantly higher levels of the 50 kDa isoform and ISG15 (2.56 and 1.7 fold increase, respectively) than their phagocytic counterparts (PKH2$^{hi}$) (Fig. 1c–e). In addition to the predicted band at 15 kDa, we also detected ISG15 immunoreactivity at the 60 kDa band with 3-fold higher level in non-phagocytic than phagocytic macrophages. This band likely corresponds to a protein modified by the ubiquitin-like nature of ISG15. Thus, non-phagocytic resolution phase macrophages exhibit increased expression of IFN-β and its downstream signaling.

Since non-phagocytic macrophages are generated during the resolution phase of inflammation, we monitored the kinetics of peritoneal fluid IFN-β during insult. We detected two peaks in IFN-β level during zymosan A-induced peritonitis in mice. The first peak occurred during the inflammatory phase (at 4 h post peritonitis initiation (PPI)), and the second one during the resolution phase (at around 72 h PPI) (Fig. 1f). Unchallenged peritoneal macrophages expressed IFN-β protein, which expression was reduced at 4 h upon exposure to zymosan A. Subsequently, at 24–72 h PPI, IFN-β levels in peritoneal macrophages exceeded the expression displayed in unchallenged macrophages (Supplementary Fig. 2f). These results would indicate production of IFN-β by peritoneal resident macrophages prior to challenge and its release in response to inflammatory stimuli. Monocytes would then infiltrate the inflamed peritoneum, differentiate to macrophages and phagocytose apoptotic PMN, eventually resulting in loss of phagocytosis and IFN-β expression during the resolving phase. To test this hypothesis, we challenged resolution phase macrophages with apoptotic cells, the pro-resolving cytokine TGF-β or the viral mimicry poly (I:C). Apoptotic cells or TGF-β evoked modest increases in IFN-β secretion, whereas poly(I:C) resulted in a robust increase (Fig. 1g). These findings suggest that progressive accumulation of apoptotic PMN by high-burden non-phagocytic macrophages during resolving inflammation leads to de novo synthesis and secretion of IFN-β and the activation of IFN-β signaling.

**IFN-β contributes to timely resolution of E.coli pneumonia.** To explore the role for IFN-β in resolving inflammation, we studied the kinetics of its production and development of PMN apoptosis in a model of spontaneously resolving bacterial pneumonia. As anticipated, intratracheal instillation of live E. coli evoked neutrophil-mediated lung injury that peaked at around 6 h and

resolved within 48 h without treatment (Fig. 2). WT mice rapidly cleared bacteria (Fig. 2a), followed by rapid decline in E. coli-induced edema (Fig. 2b, c), tissue and BAL fluid neutrophils (Fig. 2e–g) and increases in BAL fluid monocytes (Fig. 2h). These changes were associated with significant elevations in BAL fluid IFN-β level, peaking at 24 h post-E. coli instillation (Fig. 2d) parallel with increases in the percentage of apoptotic neutrophils assessed by Annexin-V staining (Fig. 2i) and the percentage of macrophages containing apoptotic bodies (Fig. 2j). We also detected a strong positive correlation between BAL fluid IFN-β levels and percentage of apoptotic neutrophils (Fig. 2k).

Next, we investigated the impact of neutralizing IFN-β activity on the resolution of E. coli pneumonia. We found that pretreatment of mice with an anti-IFN-β antibody delayed E. coli clearance at 24 h (Fig. 3a) while enhanced E. coli–evoked edema formation (Fig. 3b, c), lung myeloperoxidase content, an index of neutrophil accumulation (Fig. 3d), BAL fluid total leukocyte (Fig. 3e), and PMN (Fig. 3f) and monocyte counts (Fig. 3g) in comparison to IgG controls. By contrast, the anti-IFN-β antibody markedly reduced the percentage of apoptotic PMN in the BAL fluid at 24 h as compared to IgG (Fig. 3h). We detected slight increases in the percentage of apoptotic PMN at 48 h post E. coli, likely due to hampered efferocytosis by macrophages. Along these lines, the anti-IFN-β antibody impinged efferocytosis at both 24 h and 48 h (Fig. 3i). Thus, blocking IFN-β abrogates the resolution of inflammation by limiting PMN apoptosis and efferocytosis and preventing clearance of infiltrating PMN.

**IFN-β promotes apoptosis in inflammatory neutrophils.** To investigate the mechanisms by which IFN-β governs resolution, we next studied the impact of IFN-β on apoptosis of inflammatory neutrophils. We detected similar percentages of apoptotic PMN in the peritoneum of Ifnb$^{+/+}$ and Ifnb$^{-/-}$ mice at 4 h PPI (Fig. 4a), whereas neutrophils from Ifnb$^{-/-}$ mice displayed lower apoptosis rate than their Ifnb$^{+/+}$ counterparts following 24 h culture ex vivo and reduced inhibition of apoptosis by the caspase inhibitor Q-VD. Conversely, treatment of peritoneal PMN ex vivo with murine IFN-β led to increases in the number of annexin-V-positive cells (Fig. 4b) and cleavage of caspase 3 (Fig. 4c), indicating increased apoptosis. Furthermore, reduced peritoneal leukocyte and neutrophil numbers at 4 h PPI were observed in Ifnb$^{-/-}$ or anti-IFN-β-treated WT mice, as compared with untreated Ifnb$^{+/+}$ ones (Supplementary Fig. 3). These results indicate that IFN-β promoted resolution by enhancing the apoptosis of inflammatory neutrophils rather than by inhibiting neutrophil influx into the inflamed tissue.

To confirm that our findings in mice translate to human innate immunity, we cultured human neutrophils for 24 h with bacterial DNA (CpG DNA) in the presence of IFN-β. Confirming previous results[14], CpG DNA prolonged neutrophil survival by delaying constitutive apoptosis as indicated by reduced staining for annexin-V, prevention of the collapse of mitochondrial transmembrane potential and hypoploid nuclei (Fig. 4d–g). While IFN-β alone failed to affect neutrophil apoptosis (Supplementary Fig. 4a), pretreatment of PMN with IFN-β countered the survival cue from CpG DNA in a concentration-dependent manner (Fig. 4d–g). Likewise, when PMN were first challenged with CpG DNA and then treated with IFN-β at 60 min post-CpG DNA, the pro-apoptotic action of IFN-β was dominant over CpG DNA-generated effects (Fig. 4h–k). Notably, this inhibitory action was still detectable when IFN-β was added 120–240 min post-CpG DNA (Supplementary Fig. b, c). IFN-β attenuated CpG DNA preservation of Mcl-1 expression (Fig. 4l, m), a key regulator of life span in human PMN[33,34]. Furthermore, the pro-apoptotic

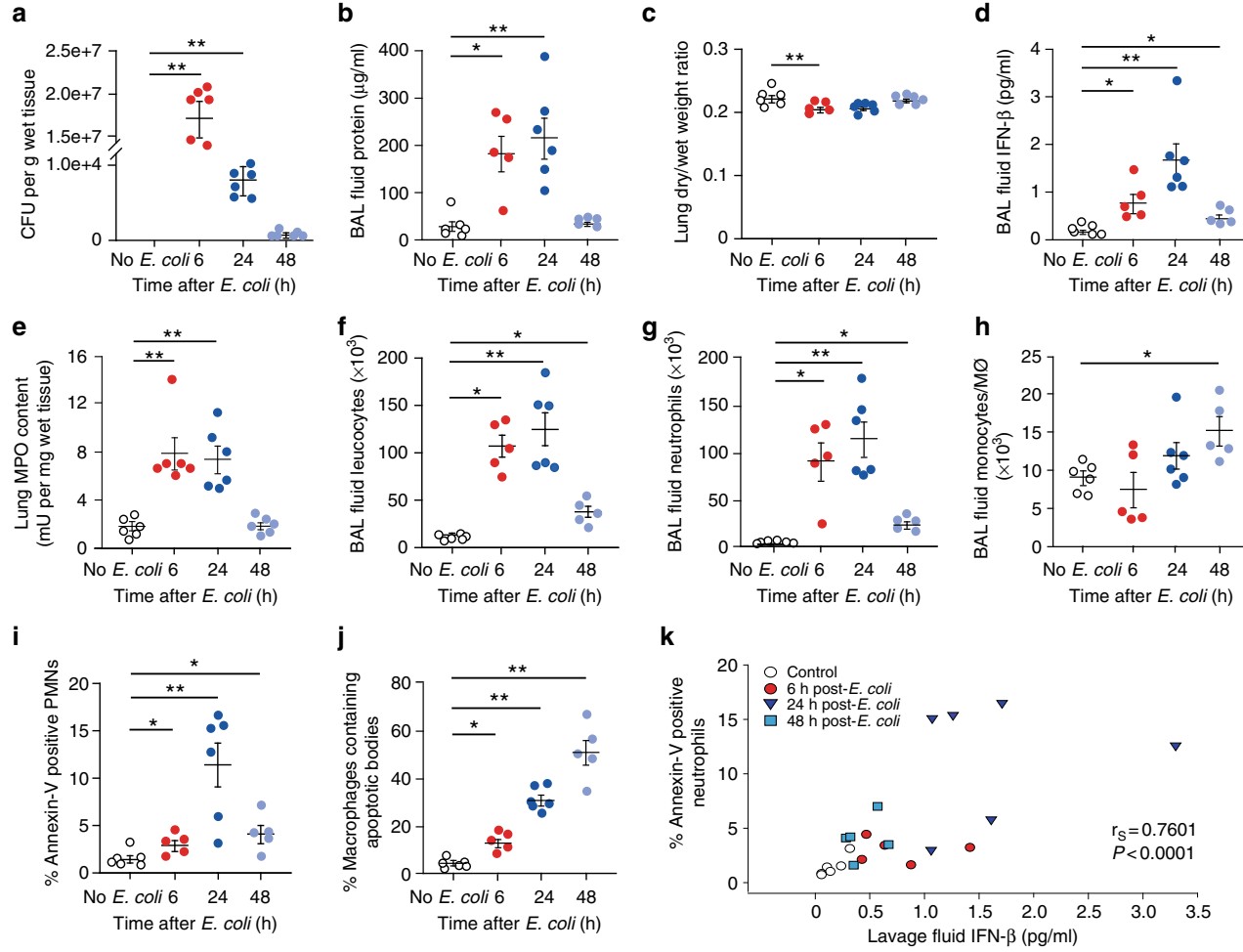

**Fig. 2** Enhanced IFN-β production during the resolution of *E.coli* pneumonia. Under isoflurane anesthesia, female C57BL/6 mice were injected intratracheally with 5*10[6] live *E. coli*. At the indicated times, lungs were collected without lavage and analyzed for *E. coli* content (**a**), lung dry-to-wet weight ratio (**c**) and tissue MPO activity (**e**). In separate groups of mice bronchoalveolar lavage fluid protein concentration (**b**), IFN-β levels (**d**), total leukocyte (**f**), neutrophil (**g**) and monocyte/macrophage numbers (**h**) and the percentage of annexin-V-positive (apoptotic) PMN (**i**) and the percentage of BAL fluid macrophages containing apoptotic bodies (**j**) were determined. Results are means ± SEM ($n = 6$ mice per group). *$P < 0.05$, **$P < 0.01$ (Dunn's multiple contrast hypothesis test). **k** Neutrophil apoptosis positively correlates with lavage fluid IFN-β levels (Spearman correlation analysis). Source data are provided as a Source Data file

action of IFN-β was also dominant over survival cues generated by LPS or the acute-phase protein serum amyloid A (Supplementary Fig. 5a–p). By contrast, IFN-β failed to affect PMN apoptosis induced by the phagocytosis of *E. coli* (Supplementary Fig. 5q, r). Thus, IFN-β can efficiently antagonize survival signals and facilitate apoptosis in activated human neutrophils, but not under homeostatic settings.

**IFN-β drives neutrophil apoptosis through STAT3**. To investigate the signaling events underlying IFN-β promotion of apoptosis in human PMN, we probed cytosolic and nuclear fractions of cell lysates with antibodies against phospho-STAT1 or phospho–STAT3. We detected significant increases in the levels of phosphorylated STAT1 and STAT3 both in the cytosolic or nuclear fractions of IFN-β-exposed PMN (Fig. 5a, b), whereas CpG DNA did not affect phosphorylation of STAT1 or STAT3. Next, we assessed the contribution of STAT signaling to the pro-apoptotic effect of IFN-β. Preincubation of PMN with the STAT3 inhibitor WP1066, but not the STAT1 inhibitor fludarabine, efficiently blocked the pro-apoptotic action of IFN-β in PMN challenged with CpG DNA (Fig. 5c–f).

CpG DNA reduced surface expression of IFNαR1 that binds IFN-β within 2 h of culture (Supplementary Fig. 6a–b), whereas IFN-β did not affect the uptake of labeled oligodeoxynucleotides by PMN (Supplementary Fig. 6c). These findings indicate the ability of IFN-β to redirect neutrophils to apoptosis even when its receptor is slightly downregulated in the presence of CpG DNA, and suggest that diminishing IFNαR1 expression is part of the survival cascade induced by CpG DNA. The IFN-β pro-apoptotic signal appears to be predominantly mediated through STAT3 activation and not by reducing CpG DNA uptake by human PMN.

**IFN-β enhances macrophage efferocytosis**. Having documented enhanced IFN-β generation by non-phagocytic macrophages and IFN-β promotion of PMN apoptosis, next we tested whether it could promote efferocytosis. We found that macrophages from *Ifnb*[−/−] mice contained lower amounts of apoptotic nuclei and engulfed on average significantly lower numbers of apoptotic PMN than macrophages from *Ifnb*[+/+] mice (Fig. 6a, b). Notably, *Ifnb*[−/−] mice contained 1.85-fold higher numbers of macrophages that did not engulf PMN at all than *Ifnb*[+/+] mice (Fig. 6c).

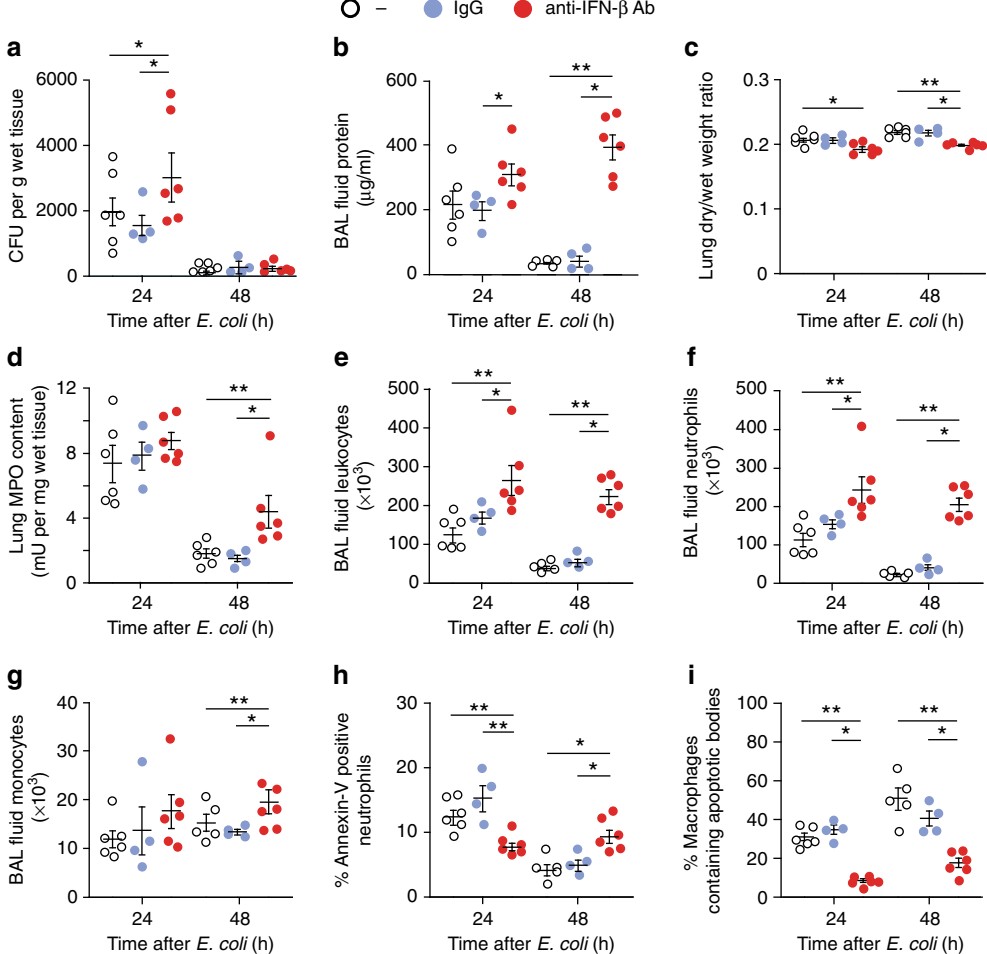

**Fig. 3** IFN-β is essential for the resolution of *E. coli* pneumonia. Female C57BL/6 mice were injected intratracheally with 5*10⁶ live *E. coli* with anti-IFN-β antibody (1 μg/20 g b.w.) or isotype control (IgG) or saline ("-"). At 24 or 48 h, lungs were removed without lavage and analyzed for *E. coli* content (**a**), lung dry-to-wet weight ratio (**c**) and tissue MPO activity (**d**). In separate groups of mice bronchoalveolar lavage fluid protein concentration (**b**), total leukocyte (**e**), neutrophil (**f**) and monocyte/macrophage numbers (**g**), the percentage of annexin-V-positive (apoptotic) PMN (**h**), and the percentage of macrophages containing apoptotic bodies (**i**) were determined. Results are means ± SEM (*n* = 6 mice per group for control and anti-IFN-β, or 4 for IgG). *$P < 0.05$, **$P < 0.01$ (Dunn's multiple contrast hypothesis test). Source data are provided as a Source Data file

Since monocyte/macrophage migration to and from the resolving site may confound detection of efferocytosis in vivo[9], we evaluated the direct uptake of CypHer-loaded apoptotic cells by resolution phase *Ifnb*⁻/⁻ macrophages ex vivo (Fig. 6d, e shows representative results). We detected significant decreases in apoptotic cell uptake by *Ifnb*⁻/⁻ macrophages and in the percentage of efferocytosing macrophages (Fig. 6e–g). Importantly, the reduction in efferocytosis was associated with increased motile morphology as *Ifnb*⁻/⁻macrophages appeared to be more spread and activated, and had more extensive filopodia (Fig. 6e, arrowheads). Of note, the reduction in efferocytosis in *Ifnb*⁻/⁻ macrophages was not due to changes in macrophage origin or number. Indeed, the recovered peritoneal macrophage population was still dominated by Tim4⁻ macrophages (Supplementary Fig. 1a, b) and macrophage numbers were in fact reduced in *Ifnb*⁻/⁻ mice at 48 h PPI (Fig. 6h), whereas the number of peritoneal resident macrophages was similar in unchallenged wild type and *Ifnb*⁻/⁻ mice (Supplementary Fig. 1a–c). These results would suggest an important role for IFN-β in attracting monocytes to inflamed sites, as previously reported in pristane-evoked inflammation[27]. Taken together, these findings indicate that IFN-β enhances efferocytosis by resolution phase macrophages in vivo and ex vivo.

**IFN-β promotes reprogramming of resolution phase macrophages.** The uptake of apoptotic cells by macrophages leads to reduced secretion of pro-inflammatory cytokines and increased secretion of anti-inflammatory and pro-resolving mediators when stimulated by microbial moieties[35–37]. This process is termed reprogramming and is critical for the prevention of autoimmunity and the restoration of tissue architecture and function[4,7]. Since *Ifnb*⁻/⁻ macrophages are less efferocytic and more activated than their *Ifnb*⁺/⁺ counterparts, we investigated whether reprogramming is hampered in these macrophages. We found that resolution phase *Ifnb*⁻/⁻ macrophages secrete significantly lower amounts of IL-10 (55% decrease) and significantly higher amounts of IL-6, IL-12 and CCL3 (10%, 18%, and 120% increases, respectively) upon LPS stimulation than their *Ifnb*⁺/⁺ counterparts (Fig. 7a–d). Furthermore, *Ifnb*⁺/⁺ macrophages significantly upregulated IL-10 (183% increase) and diminished IL-12 secretion (73% reduction) in response to IFN-β, but not IFN-α ex vivo (Fig. 7e–f). Notably, peritoneal resident macrophages from *Ifnb*⁻/⁻ mice did not show reduced reprogramming. Rather, these macrophages secreted significantly reduced levels of either IL-12 and TNFα or IL-10 in comparison to their *Ifnb*⁺/⁺ counterparts (Supplementary Fig. 1d-f). Thus, IFN-β directly promotes the reprogramming of monocyte-derived resolution phase macrophages.

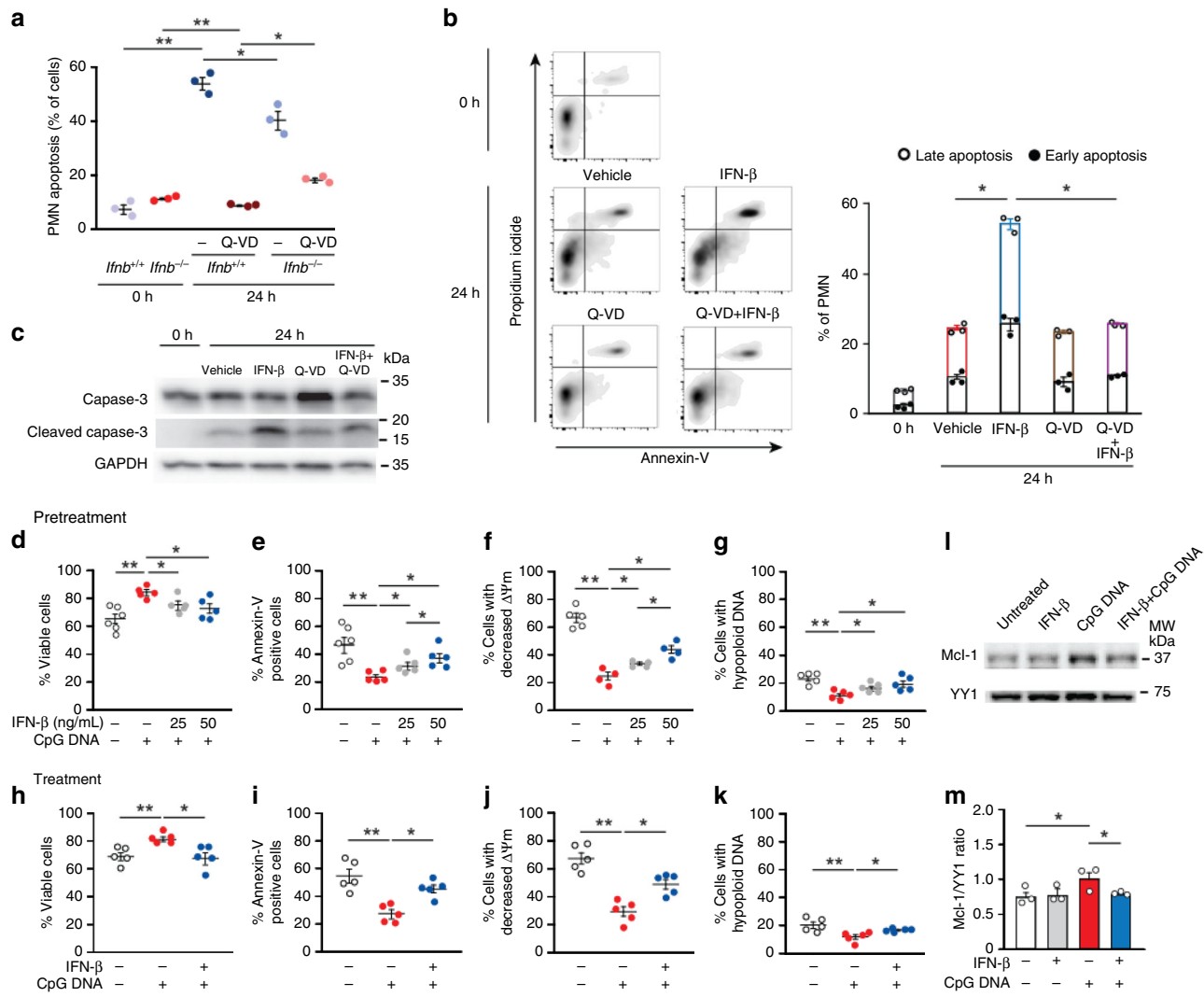

**Fig. 4** IFN-β promotes apoptosis in mouse and human neutrophils. **a** Peritoneal PMN were recovered from male *Ifnb*[+/+] or *Ifnb*[−/−] mice at 4 h PPI and stained immediately with annexin-V and propidium iodide to assess apoptosis and cell viability, respectively with flow cytometry. Alternatively, cells were cultured with or without the pan-caspase inhibitor Q-VD (10 μM) for 24 h and then assessed for apoptosis. Results are mean ± SEM from three independent experiments. *P < 0.05, **P < 0.01 (Tukey's HSD). **b**, **c** Peritoneal PMN were recovered from *Ifnb*[+/+] mice 24 h PPI and cultured ex vivo with IFN-β (20 ng/ml) and/or Q-VD (10 μM) for 24 h. Apoptosis was evaluated as above. In some experiments, PMN lysates were prepared after 6 h culture and immunoblotted for cleaved (active) caspase-3. Results are representatives for three independent experiments. *P < 0.05 (Tukey's HSD). **d**–**g** Human PMN (5 × 10⁶ cells/ml) were pretreated with human recombinant IFN-β (25–50 ng/ml) for 10 min and then challenged with CpG DNA (1.6 μg/ml) or (**h**–**k**) first challenged with CpG DNA (1.6 μg/ml) and then treated with IFN-β (50 ng/ml) at 60 min post-CpG DNA. Cell viability (**d**, **h**), annexin-V staining (**e**, **i**), mitochondrial transmembrane potential (ΔΨm; CMXRos staining, **f**, **j**) and nuclear DNA content (**g**, **k**) were analyzed after culturing neutrophils for 24 h with CpG DNA. Results are mean ± SEM of 5 experiments with different blood donors. *P < 0.05, **P < 0.01 (Dunn's multiple contrast hypothesis test). **l**, **m** Human PMN lysates, prepared following 4 h culture with IFN-β (50 ng/ml) with or without CpG DNA, were immunoblotted for Mcl-1 or the ubiquitous transcription factor YY1 as a loading control. Representative blots (**l**) and densitometry analyses (**m**) for three independent experiments. *P < 0.05, **P < 0.01 (Dunn's multiple contrast hypothesis test). Source data are provided as a Source Data file

The uptake of apoptotic PMN and pro-resolving mediators promote macrophage loss of phagocytosis and consequently their conversion from the CD11b[high] to CD11b[low] phenotype hallmarked by reduced expression of the arginase 1 and increased expression of 12/15-LO[9,10]. Our results indicate that IFN-β also facilitates these processes either ex vivo by increasing the percentage of CD11b[low] macrophages (Fig. 7g), or in vivo by reducing arginase 1 expression and upregulating 12/15-LO expression (Fig. 7h–i).

**IFN-β accelerates the resolution of bacterial inflammation.** Pro-resolving lipid mediators, such as RvD1 and RvD5, have been

shown to enhance *E. coli* clearance and to promote the resolution of bacterial inflammation in vivo[38]. Having shown that genetic deletion or neutralization of IFN-β limited various resolution-promoting indices, we examined whether treatment with exogenous IFN-β could accelerate the resolution of inflammation. Treatment of mice with IFN-β at the peak of inflammation resulted in accelerated clearance of *E. coli* at 24 h (Fig. 8a), marked reductions in edema (Fig. 8b, c), lung myeloperoxidase content (Fig. 8d), BAL fluid total leukocytes (Fig. 8e), and PMN counts (Fig. 8f) as compared with vehicle control. Furthermore, IFN-β also enhanced monocyte/macrophage numbers in the lungs at 24 h post-*E. coli*, increased the percentage of apoptotic

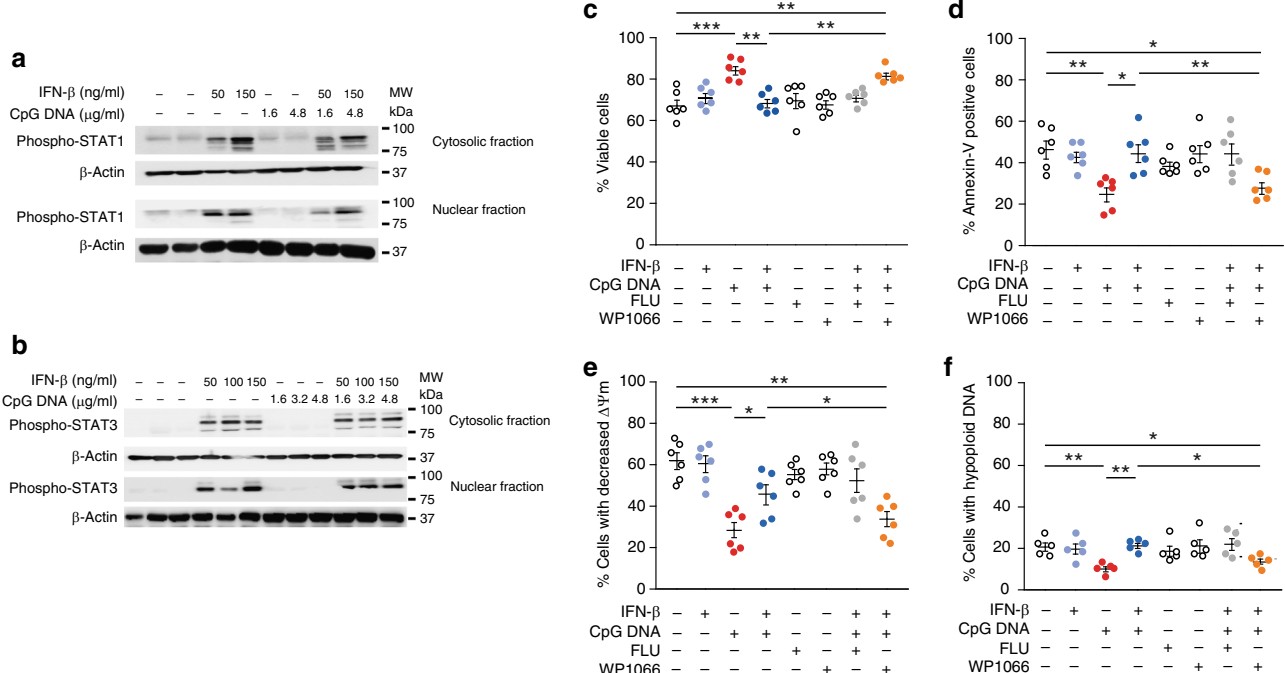

**Fig. 5** IFN-β promotes neutrophil apoptosis through STAT3 activation. **a, b** Human PMN (5*10[6] cells/ml) were pretreated with human IFN-β (50–150 ng/ml) for 10 min and then challenged with CpG DNA for 30 min. The cells were then lysed, cytosolic and nuclear fractions were prepared and immunoblotted for phospho-STAT1 (**a**) or phospho-STAT3 (**b**). β-actin served as a loading control. Blots are representative for 3 independent experiments. **c–f** Human PMN were pre-incubated with the STAT1 inhibitor fludarabine (FLU; 25 μM) or the STAT3 inhibitor WP1066 (5 μM) for 30 min before addition of human recombinant IFN-β (50 ng/ml) for 10 min and then challenged with CpG DNA. After culture for additional 24 h, cell viability (**c**), annexin-V staining (**d**), mitochondrial transmembrane potential (ΔΨm, CMXRos staining) (**e**) and nuclear DNA content (**f**) were analyzed. Results are mean ± SEM of 6 experiments with different blood donors. *P < 0.05, **P < 0.01, ***P < 0.001 (Dunn's multiple contrast hypothesis test). Source data are provided as a Source Data file

PMN and macrophages containing apoptotic PMN at 24 and 48 h post-*E. coli*. Likewise, in the peritonitis model, treatment with exogenous IFN-β enhanced efferocytosis assessed both in vivo and ex vivo and rescued hampered efferocytosis in *Ifnb*[−/−] macrophages (Supplementary Fig. 7a, b). Moreover, exogenous IFN-β facilitated macrophage reprogramming in LPS-stimulated WT mice and restored reprogramming in *Ifnb*[−/−] macrophages, at least in part, as evidenced by increased IL-10 secretion with concomitant reduction in IL-12, IL-6, TNFα, and CCL3 secretion (Supplementary Fig. 7c–g). Thus, exogenous IFN-β acts as a bona fide pro-resolving mediator in vivo by accelerating bacterial clearance, enhancing PMN apoptosis and efferocytosis, and macrophage reprogramming.

**Irreversible loss of phagocytosis blunts efferocytic modules.** Having shown that macrophages that lost their phagocytic capacity are key effectors in resolving inflammation and the contribution of IFN-β to this process, we next characterized the properties of non-phagocytic macrophages, in particular with respect to the uptake of apoptotic cells. We found that the phagocytic capacity of F4/80[+] macrophages (determined by the uptake of PKH2-PCL[−] green) was initially increased during the early phase of resolution of peritonitis (24–48 h) followed by significant reductions at 48–72 h PPI (Fig. 9a) without detectable changes in the percentage of phagocytic macrophages (Fig. 9b). Thus, the emergence of non-phagocytic macrophages was indeed due to a loss of phagocytic properties rather than to a loss of the phagocytic macrophage population.

Next, we examined whether various targets, like apoptotic cells or latex beads (LB) or IgG-opsonized LB labeled with green fluorescence alongside PKH2-PCL red into the

peritoneal cavity during the resolving phase of peritonitis, and assessed the uptake of PKH2-PCL and labeled targets by macrophages. Our results show that only macrophages that engulfed high amounts of PKH2 were able to engulf LB or IgG-opsonized LB (Fig. 9c–d), moreover these PKH2[hi] macrophages phagocytosed more apoptotic cells than their PKH2[lo] counter-parts, while PKH2[−] macrophages did not engulf apoptotic cells at all (Fig. 9e). Thus, PKH2 uptake correlates well with the phagocytosis of target particles, and the initial apoptotic cell engulfment seem to limit both efferocytosis and PKH2 acquisition. Along these lines, we also found that only apoptotic cells diminished PKH2 uptake by macrophages (36% reduction), whereas LB or opsonized LB did not affect uptake (Fig. 9f–g). Since uptake of high numbers of apoptotic cells parallels with loss of phagocytosis, it is conceivable that its termination will lead to reversal of non-phagocytic macrophages to the phagocytic state. To address whether loss of phagocytosis is reversible, we first injected PKH2-green into mice undergoing peritonitis followed by an injection of PKH2-red 12 h later. Peritoneal macrophages were collected and analyzed for dye uptake at 4–8 h after the second injection (Fig. 9h), and PKH2-green[+] macrophages were phenotyped according to the dye ratio. Thus, macrophages that acquired equal amounts of PKH2-green and PKH2-red (the diagonal group in the dot-plot) represent sustained phagocytosis and constituted the majority of macrophages (Fig. 9i). Macro-phages that engulfed more PKH2-red than PKH2-green represent increased phagocytosis and constituted a minute fraction of the population. Macrophages that engulfed more PKH2-green than PKH2-red represent reduced phagocytosis and the percentage of this subset increased with time while sustained phagocytosis was reduced (Fig. 9i, j show representative images and cumulative data, respectively). Our transcriptomic analysis revealed markedly

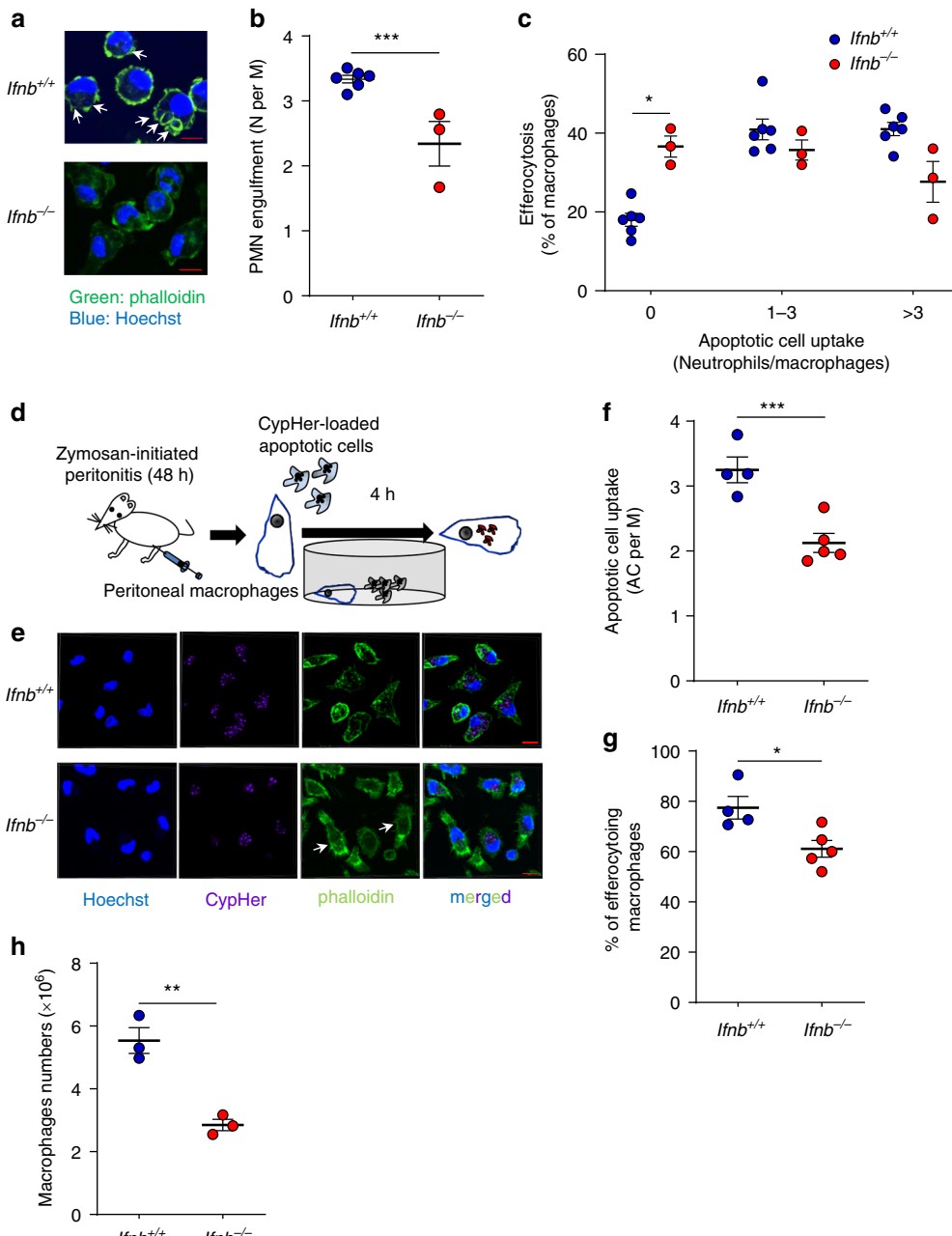

**Fig. 6** IFN-β enhances macrophage efferocytosis during the resolution of inflammation. **a–c** Peritoneal macrophages were recovered from male *Ifnb*[+/+] or *Ifnb*[−/−] mice at 48 h PPI and stained with Hoechst 33342 and FITC-phalloidin (**a**). Fluorescent images of select frames were taken using high-resolution microscopy achieved by 3D confocal (z-stack) scanning with a Nikon A1-R confocal fluorescent microscope. The number of apoptotic PMN nuclei in each macrophage were enumerated (see arrowheads for illustration) using the Nikon NIS-Elements microscope imaging software and average engulfment per macrophage (**b**) or engulfment according to thresholds (**c**) were calculated. Representative images (**a**) and means ± SEM (**b, c**) for six (*Ifnb*[+/+]) and three (*Ifnb*[−/−]) experiments. **d–f** Peritoneal macrophages were recovered from male *Ifnb*[+/+] or *Ifnb*[−/−] mice at 48 h PPI and incubated with CypHer-labeled apoptotic Jurkat cells at a ratio of 1:3. After 4 h, unbound cells were washed and macrophages were stained with Hoechst 33342 and FITC-phalloidin (as illustrated in **d**). Select images of each staining and merged images are shown (**e**). The number of engulfed apoptotic cells (AC) in each macrophage (M) were counted and average apoptotic cell uptake (**f**) and the percentage of efferocytosing macrophages (**g**) were calculated. **h** The total numbers of F4/80[+] peritoneal macrophages was detected by flow cytometry and calculated. Representative images (**e**) and means ± SEM (**f–g**) for four (*Ifnb*[+/+]) and five (*Ifnb*[−/−]) independent experiments. *$P < 0.05$, ***$P < 0.005$ (Student's *t* test). Source data are provided as a Source Data file

reduced expression of the genes coding for the efferocytic receptors Mertk and CD206 with slight decreases in CD36 in non-phagocytic macrophages (6.7, 2.03, and 1.46 fold reduction, respectively). Consistently, we detected significantly lower surface expression of all three efferocytic receptors (57%, 64%, and 49% reduction for CD36, CD206, and Mertk, respectively) whereas

CD45 expression was unaffected (Fig. 9k–l). This expression profile resembles that of eosinophils that are non-phagocytic cells (Fig. 9l). Thus, macrophage loss of phagocytosis seems to be a progressive and irreversible process in vivo and is associated with reduced expression of efferocytic receptors through transcriptional and none-transcriptional mechanisms.

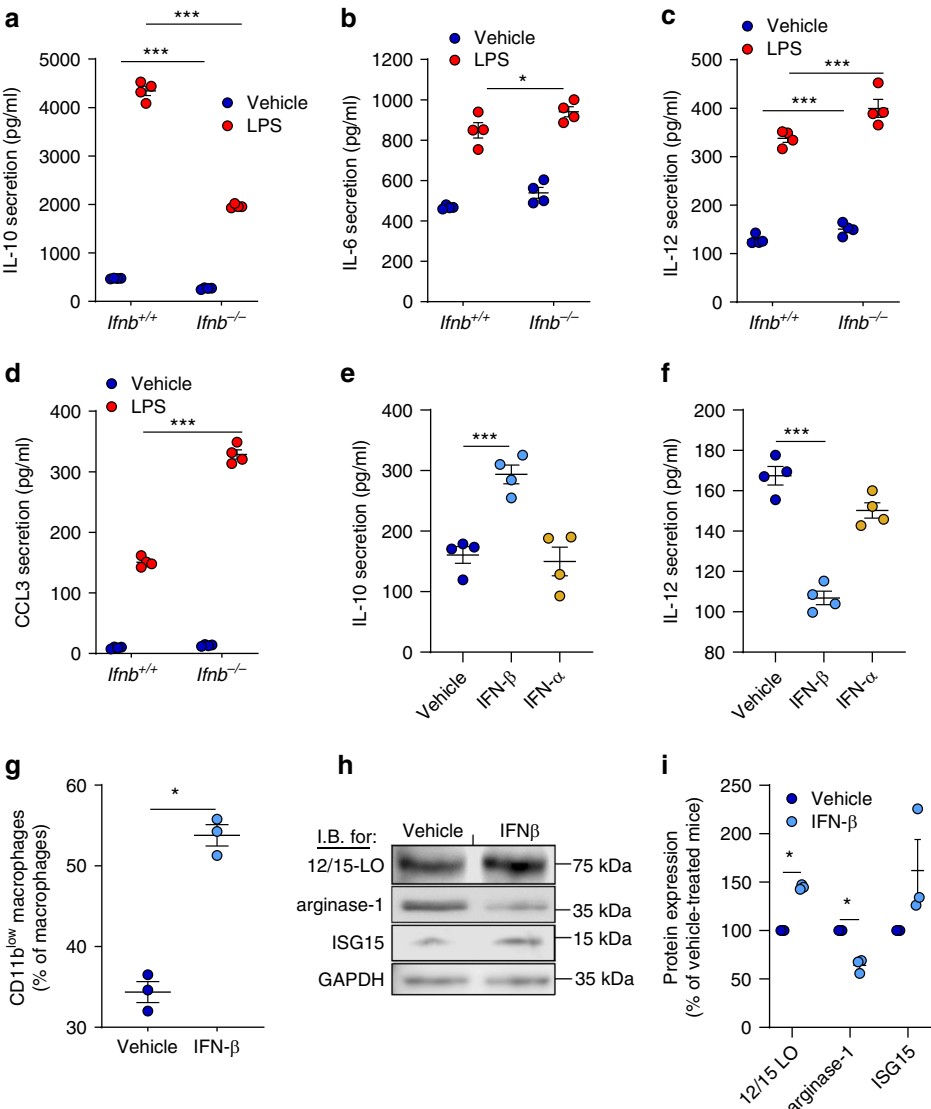

**Fig. 7** IFN-β favors macrophage reprogramming during the resolution of inflammation. **a–d** Macrophages were recovered from peritoneal exudates of male *Ifnb*^+/+^ or *Ifnb*^−/−^ mice at 48–66 h PPI and cultured with LPS (1 μg/ml) for 24 h. Culture supernatants were then collected and levels of IL-10 (**a**), IL-6 (**b**), IL-12 (**c**) and CCL3 (**d**) were determined by selective ELISAs. Results are means ± SEM from four independent experiments. *P < 0.05, ***P < 0.005 (Tukey's HSD). **e–f** Macrophages were recovered from peritoneal exudates of *Ifnb*^+/+^ mice at 48–66 h PPI and cultured with mouse IFN-β or IFN-α (20 ng/ml each) for 48 h. Then, culture supernatants were collected and levels of IL-10 (**e**) and IL-12 (**f**) were determined by ELISA. Results are means ± SEM (n = 4). ***P < 0.005 (Tukey's HSD). **g** Macrophages were recovered from peritoneal exudates of *Ifnb*^+/+^ mice 48–66 h PPI and incubated with IFN-β (20 ng/ml) for 48 h. The cells were then immunostained for F4/80 and CD11b and the percentage of CD11b^low^ macrophages was determined by flow cytometry. Results are means ± SEM from three independent experiments. ***P < 0.005 (Tukey's HSD). **h–i** Mice undergoing peritonitis were treated with IFN-β (20 ng/mouse, i.p.) or vehicle at 24 h PPI. Peritoneal macrophages were collected at 48 h PPI, lysed and immunoblotted for 12/15-LO, arginase 1, ISG15 and GAPDH. Representative blots (**h**) and densitometry analysis (means ± SEM) (**i**) for three independent experiments. *P < 0.05 (Student's *t* test). Source data are provided as a Source Data file

## Discussion

We demonstrate here a previously unappreciated role for IFN-β; upon its induction and secretion by non-phagocytic resolution phase macrophages, IFN-β promotes neutrophil apoptosis and efferocytosis as well as polarization of macrophages towards a pro-resolving phenotype, thereby facilitating timely resolution of bacterial inflammation. We propose that IFN-β mediates both feedback and bidirectional crosstalk between non-phagocytic macrophages, phagocytic macrophages and neutrophils, thereby defining a novel resolution circuit.

The role of specialized lipid mediators (SPM), like lipoxins, resolvins, protectins and maresins[12] and proteins, such as annexin A1 and galectin-1[39–41] in limiting both inflammation

and fibrotic tissue repair has been established in recent years. However, up to date, only 2 major cytokines have been associated with resolving inflammation, namely TGF-β and IL-10. TGF-β is produced during the resolution phase of spontaneously resolving inflammation in experimental animals[10,42], and specifically following the uptake of apoptotic cells by macrophages[43]. IL-10 exerts paramount anti-inflammatory and anti-fibrotic activities[44], albeit it is not always produced during the resolution phase of inflammation[42]. A common feature of both TGF-β and IL-10 is the enhancement of clearance of apoptotic PMN via efferocytosis that is essential for prevention of chronic inflammation and autoimmunity[45] and limiting the production of pro-inflammatory cytokines[43,46]. Here we show that IFN-β also

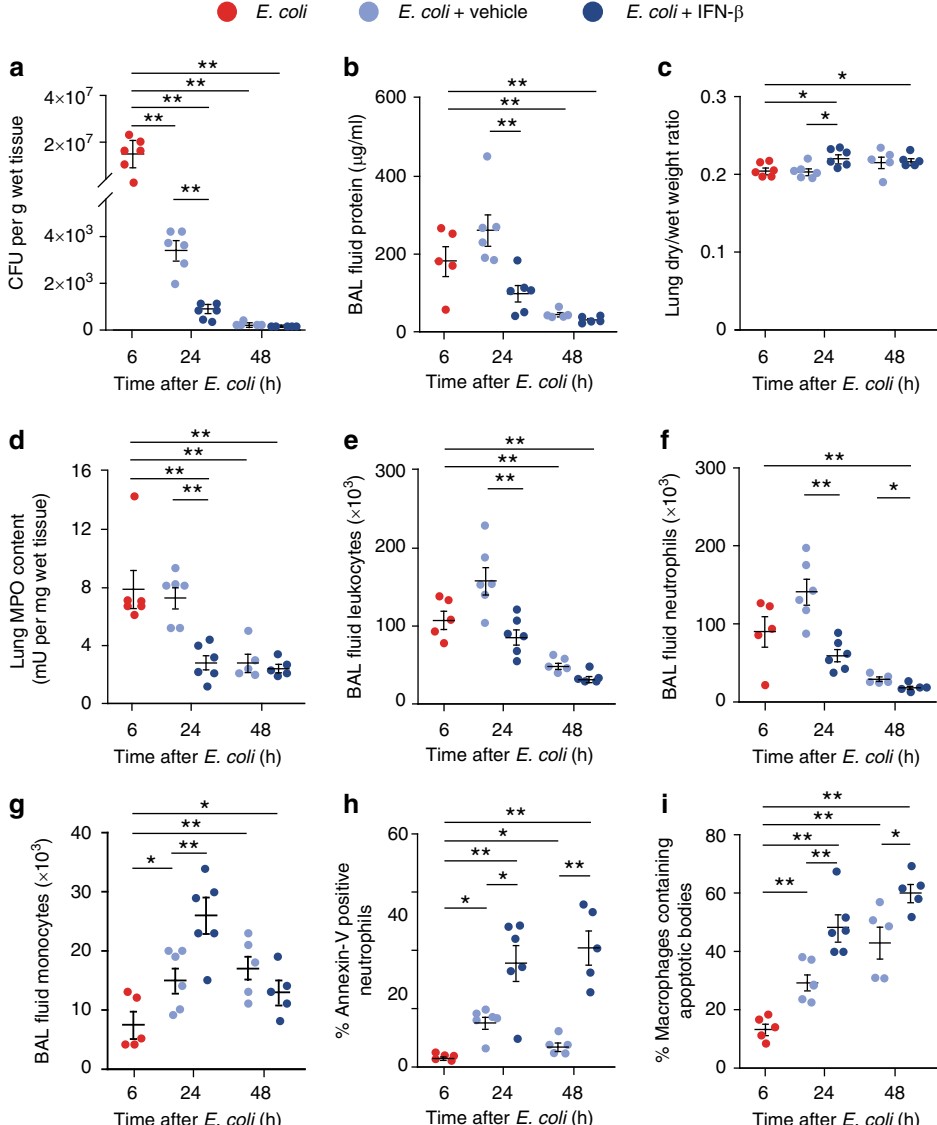

**Fig. 8** IFN-β treatment accelerates the resolution of *E. coli* pneumonia. Female C57BL/6 mice were injected intratracheally with 5*10[6] live *E. coli*. Six hours later (at the peak of inflammation), they were treated with mouse recombinant IFN-β (50 ng/20 g b.w., intraperitoneally) or vehicle. At 24 or 48 h post-*E. coli* instillation, lungs were removed without lavage and analyzed for *E. coli* content (**a**), lung dry-to-wet weight ratio (**c**) and tissue MPO activity (**d**). In separate groups of mice bronchoalveolar lavage fluid protein concentration (**b**), total leukocyte (**e**), neutrophil (**f**) and monocyte/macrophage numbers (**g**), the percentage of annexin-V-positive (apoptotic) PMN (**h**), and the percentage of macrophages containing apoptotic bodies (**i**) were determined. Results are means ± SEM (*n* = 6 mice per group for 6 and 24 h and 5 mice per group for 48 h). *P < 0.05, **P < 0.01 (Dunn's multiple contrast hypothesis test). Source data are provided as a Source Data file

possesses similar properties. Previous studies have shown that intracellular annexin A1 facilitates TLR3 and TLR9-mediated IFN-β production by macrophages[47], and IFN-β mediates the expression of the pro-resolving receptor FPR2 on macrophages[48]. Our RNA-Seq analysis (BioProject accession No: PRJNA450293) also indicate upregulated annexin A1 and FPR2 expression in non-phagocytic resolution phase macrophages (1.93 and 5.5. fold increase, respectively), presumably due to stimulation by engulfed apoptotic PMN and/or IFN-β. Hence, TLR-mediated sensing of apoptotic cell constituents might trigger an IFN-β-associated response in resolution phase macrophages.

Previous studies have shown that the uptake of apoptotic cells by phagocytes can initially enhance high-burden efferocytosis[49], which will be limited when their oxidative stress and metabolic constraints dominate[50]. We have previously shown that a subset of CD11b[low] macrophages that appear during the resolution

phase of inflammation contain high numbers of apoptotic cells but are devoid of efferocytic activity and PKH2 acquisition[9]. Hence, these macrophages were termed satiated macrophages. Whether the uptake of apoptotic cells is the sole driving force of satiation is difficult to determine, though it is well-established that macrophage reprogramming to anti-inflammatory phenotypes[2] and probably to the CD11b[low] phenotype as well[9] is mediated by efferocytosis in vivo. Hence, uptake of apoptotic cells, predominantly neutrophils, by macrophages during the resolution phase likely plays a paramount role in macrophage loss of phagocytosis and satiation. Our results demonstrate that the non-phagocytic phenotype in resolution phase macrophages exhibit increased transcription and secretion of IFN-β but not other type I IFNs. Consistently, we detected higher IFN-β levels in resolving exudates of the peritoneum and bronchoalveolar space parallel with increased rates of apoptotic death in infiltrating PMN.

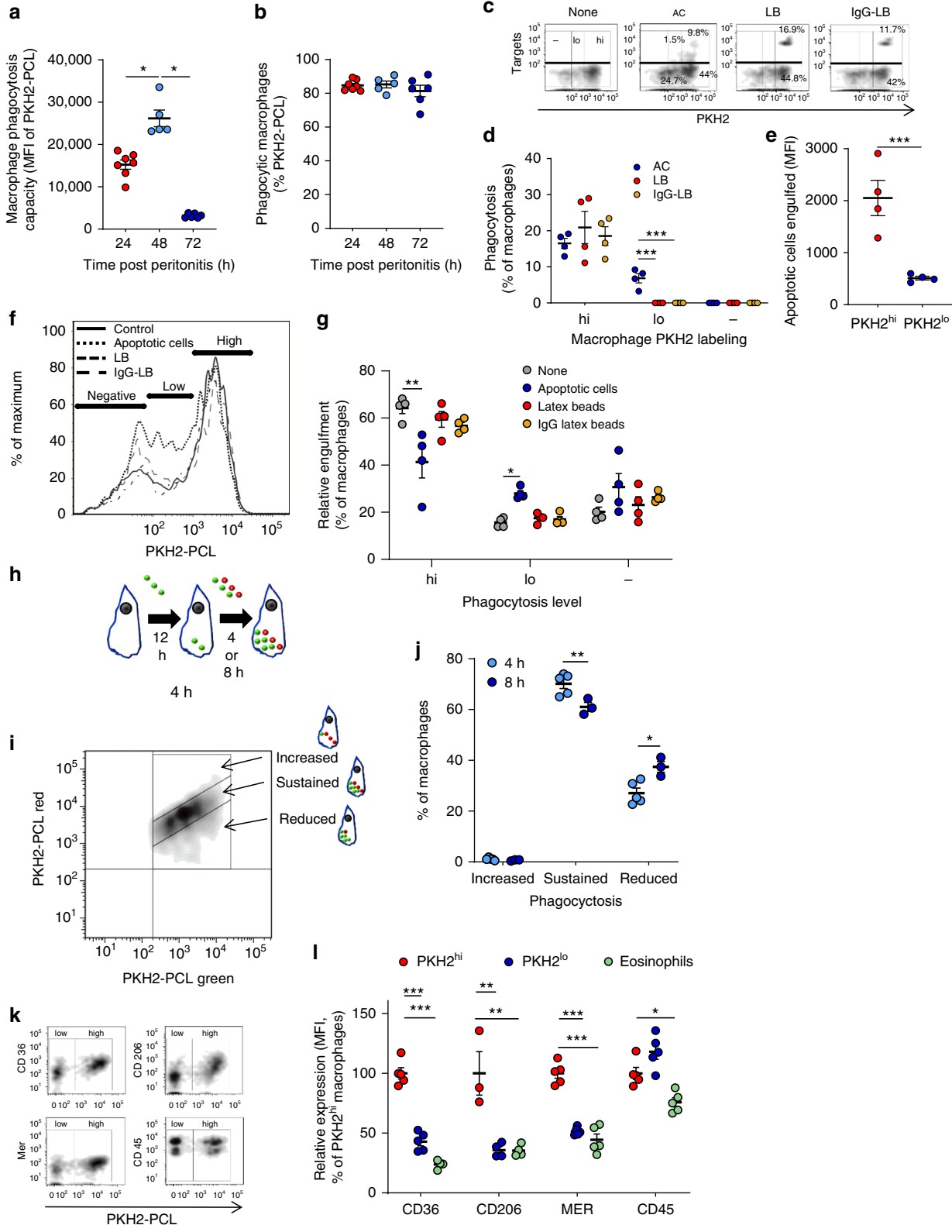

Importantly, genetic deletion or neutralization of IFN-β lead to reduced PMN apoptosis in vivo as well as upon culture ex vivo. Conversely, treatment with exogenous IFN-β enhanced neutrophil apoptosis both in the *E. coli* pneumonia and peritonitis models. It is important to note that IFN-β effectively overrode survival cues from TLR ligands, such as CpG DNA and LPS, as well as the acute-phase protein serum amyloid A, and redirected human PMN to apoptosis. Such an inhibitory action was still detectable when IFN-β was added at 4 h post-challenge, further highlighting the therapeutic potential of this cytokine. Our results point toward STAT3 activation and increased Mcl-1 degradation as main mechanisms underlying the apoptosis-promoting action of IFN-β in human PMN. Inflammatory activation of PMN seems essential for the pro-apoptotic activity of IFN-β. These findings

**Fig. 9** Characteristics of macrophages following loss of phagocytosis. **a, b** The phagocyte-specific dye PKH2-PCL green was injected I.P. to male WT mice undergoing peritonitis for 20, 44 and 68 h. After 4 h, the peritoneal cells were recovered and immuno-stained for F4/80 and CD11b. PKH2-PCL green acquisition by CD11b[high] macrophages was determined by flow cytometry. Results are means ± SEM ($n = 8$ mice for 24 h, 6 mice for 48 h, and 7 mice for 72 h) showing MFI (**a**) and the percentage of PKH2[+] cells (**b**). *$P < 0.05$ (Tukey's HSD). **c–g** Mice were injected i.p. with $3 \times 10^6$ apoptotic Jurkat cells, latex beads, IgG-opsonized latex beads or vehicle together with PKH2-PCL red at 62 h PPI. 4 h later, the peritoneal cells were recovered, immuno-stained for F4/80 and CD11b and F4/80[+] macrophages were analyzed by flow cytometry for target particle uptake and/or PKH2 engulfment (designated as PKH2-high, -low and -negative populations). Results are representative for three experiments presented as density or histogram plots (**c, f**) and means ± SEM from 4 independent experiments of % of phagocytic macrophages (**d**), particles engulfed (**e**), or PKH2 uptake (**g**). *$P < 0.05$, **$P < 0.01$ (Tukey's HSD). **h, i** PKH2-PCL green was injected I.P. to mice at 44 h PPI followed by an injection of PKH2-PCL red at 58 h PPI. Peritoneal cells were recovered at 62 or 66 h PPI and immuno-stained for F4/80 and CD11b and analyzed with flow cytometry (as illustrated in **h**). PKH2-PCL green vs. red acquisition was determined in F4/80[+] macrophages and macrophages that displayed increased, sustained or reduced phagocytosis (representative dot-plot is shown in **i**). **j** Data are means ± SEM ($n = 5$ mice for 4 h and 3 mice for 8 h). *$P < 0.05$, **$P < 0.01$ (Student's $t$ test). **k, l** PKH2-PCL red was injected I.P. to mice at 62 h PPI. 4 h later the peritoneal cells were recovered, immuno-stained for CD36, CD206 or MER, receptors involved in apoptotic cell uptake. Staining for CD45 served as a control. Receptor expression on PKH2-PCL-high (PKH2[hi]) or PKH2-PCL-low (PKH2[lo]) macrophages was determined by flow cytometry. Results are representative from $n = 4$ mice presented as dot plots (**k**) and means ± SEM relative MFI normalized to PKH2[hi] macrophage expression (**l**). *$P < 0.05$, **$P < 0.01$, ***$P < 0.005$ (Student's $t$ test). Source data are provided as a Source Data file

are in line with a previous report of IFN-β promotion of apoptosis in tumor-infiltrating neutrophils[25]. Notably, increased IFN-β levels were also detectable during the onset of inflammation in our peritonitis and pneumonia models. However, IFN-β did not abrogate neutrophil recruitment to the inflamed peritoneum. Thus, the pro-resolving actions of IFN-β on neutrophils can primarily be attributed to enhancing apoptosis and its consequences.

Our results demonstrate IFN-β promotion of efferocytosis of apoptotic PMN in vivo during the resolution phase of peritonitis and pneumonia. Importantly, this pro-efferocytic activity was independent of enhancing PMN apoptosis since the uptake of apoptotic cells ex vivo was also abrogated in $Ifnb^{-/-}$ macrophages. The findings that both TGF-β and apoptotic cells can induce IFN-β secretion by resolution phase macrophages would raise the intriguing possibility that IFN-β may, at least in part, mediate the pro-efferocytic activity of TGF-β, which is also produced during efferocytosis. We also show that IFN-β is involved in the expression of IL-10 by resolution phase macrophages and the reprogramming of these macrophages to a pro-resolving phenotype. Since IL-10 also promotes engulfment of apoptotic cells[51], it is conceivable that IL-10 might act downstream to or in concert with IFN-β to promote efferocytosis.

During the resolution of inflammation macrophages that have engulfed high numbers of apoptotic PMN downregulate CD11b and F4/80 expression and become CD11b[low] macrophages[9,10]. These macrophages might be converted from an M2-like profibrotic phenotype to a pro-resolving phenotype (termed Mres) that is characterized by the increased expression of the SPM producing enzyme 12/15-LO, and reduced expression of the M2 marker arginase 1[52]. Our transcriptome analysis confirms that phagocytic CD11b[high] macrophages display an increased gene profile commensurate with an M2, reparative phenotype. Functional gene annotations relevant to this designation include blood vessel development, extracellular matrix, protein kinase activity, and cell morphogenesis involved in differentiation and cell adhesion. The phagocytic nature of these macrophages is highlighted by the increased expression of genes associated with endocytosis, membrane invagination, cytoskeletal protein binding and adhesion molecules. The loss of phagocytic properties that characterizes non-phagocytic resolution phase macrophages is associated with reduced expression of these functional gene groups and an increase in the expression of a distinct IFN-β-related gene signature with functional designations like immune, defense and inflammatory responses. Other upregulated gene groups that might be related to the non-phagocytic CD11b[low]

phenotype include locomotor behavior and oxidation-reduction processes. These functions are linked to loss of phagocytic function and increased migration to remote sites, which are characteristic features of these macrophages[9]. Our results indicate that IFN-β promoted macrophage efferocytosis and reprogramming, as well as polarization towards the Mres phenotype in vivo by increasing the percentage of CD11b[low] macrophages in resolving exudates and the expression of 12/15-LO while reducing the expression of arginase 1. Previous studies have reported additional actions of IFN-β that are consistent with resolution. These include the induction of IL-10 following bacterial stimulation[24,53], limiting the lethality of LPS[23], inhibition of lung fibrosis[54] by reducing TGF-β levels, and abrogating osteoclast development and inflammatory bone destruction[55]. Based on these and our current results, we conclude that IFN-β is a paramount effector cytokine in resolving inflammation.

Since non-phagocytic macrophages are an important source of IFN-β during the resolution of inflammation, we further characterized their properties and regulation of their tissue levels. Our results suggest that non-phagocytic macrophages are formed from the phagocytic pool of macrophages that originates from blood monocytes and characterized as F4/80[hi] macrophages. As resolution progresses, the uptake of apoptotic PMN and other cells, but not other phagocytic targets, converts resolution phase macrophages to the non-phagocytic state with complete loss of phagocytic capacity. Subsequently, non-phagocytic macrophages will reduce surface expression of efferocytic receptors, like CD11b, CD36, CD206, and Mer as well as the transcripts of other efferocytosis-related genes, such as low-density lipoprotein receptor-related protein 1 (LRP1), ATP-binding cassette A1 (ABCA1), Dynamin 1 and 2, and dedicator of cytokinesis 1 (DOCK) 1 (BioProject accession No: PRJNA450293)[56]. IFN-β expression in macrophages is associated with a posttranslational modification that generates 50–66 kDa products that are stored in the secretory vesicles. These modifications need to be removed for secretion because macrophages release only the 25–35 kDa form. Additional studies on the mechanism that controls the secretion of IFN-β and possibly other cytokines as well, are undoubtedly warranted to further our understanding of the role of IFN-β. For clinical aspects of IFN-β therapy please see Supplemental Discussion.

Collectively, our findings highlight the importance of efferocytic satiation and identify a novel IFN-β−mediated circuit that contributes to the termination of bacterial inflammation through facilitating PMN apoptosis and efferocytosis, and macrophage reprogramming to pro-resolving phenotypes. These findings would also suggest the therapeutic potential of IFN-β in chronic

and non-resolving inflammation, as well as fibrotic disorders and wound repair.

## Methods

**Mice.** *Ifnb*$^{-/-}$ male and female mice (Erlandsson et al., 1998) were a kind gift from Prof. Issazadeh-Navikas (University of Copenhagen). C57BL/6 WT male and female mice were purchased from Harlan Laboratories and Charles River Laboratories, respectively. All mice were aged 8–15 weeks and did not undergo previous procedures. All mice were fully backcrossed to C57BL/6 and housed under a 12 h:12 h light-dark cycle and specific pathogen-free conditions, up to 5 mice per cage. Mice were fed standard pellet chow and reverse osmosis water ad libitum. Animal experiments were approved by the Committee of Ethics, University of Haifa (authorization no. 246/14) or the Animal Care Committee, Maisonneuve-Rosemont Hospital (protocol no. 2015/31) and mice were maintained under the respective committee's ethical regulations for animal testing and research.

**Human blood donors.** Venous blood (20 ml, anticoagulated with sodium heparin, 50 U/ml) was obtained from nonsmoking apparently healthy volunteers (male and female, 26–65 years) who had denied taking any medication for at least 2 weeks. The Clinical Research Committee at the Maisonneuve-Rosemont Hospital approved the experimental protocols (project no. 99097) and each volunteer provided written informed consent.

**Mouse peritonitis model.** Male C57BL/6 *Ifnb*$^{-/-}$ or *Ifnb*$^{+/+}$ mice were randomly assigned to experimental groups. Mice were injected I.P. with zymosan A (1 mg/ml in PBS, 1 ml per mouse). PKH2-PCL green (0.25 mM, 0.5 mL) was injected I.P. at 62 h and peritoneal exudates were collected 4 h later. Peritoneal cells were stained with PE-conjugated rat anti-mouse F4/80 (1:100, Biolegend 123110), and PerCP-conjugated rat anti-mouse CD11b (1:100, Biolegend 101230) and F4/80$^+$ macrophages were sorted using the FACSaria III sorter (Beckton-Dickinson). In separate experiments, peritoneal exudates were collected from unchallenged mice or 4, 24, 48, 66–72 or 96 h post-zymosan A and cellular content was analyzed by flow cytometry as above. In separate experiments, mice challenged with zymosan A for 42 h or unchallenged were injected intraperitoneally with pre-made liposomal clodronate (1 or 0.1 mg/mouse, respectively) or empty liposomes. Then, unchallenged mice were injected with zymosan A. After additional 24 h, peritoneal fluids were collected from all mice and analyzed by Western blotting for IFN-β as indicated below. Clodronate treatment resulted in an 84% reduction in peritoneal macrophages and no significant changes in the percentage of splenic macrophages.

**Mouse pneumonia model.** Under isoflurane anesthesia, female C57BL/6 mice were first injected I.P. with saline, rat anti-mouse IFN-β monoclonal Ab (clone 7F-D3, Abcam) or isotype-matched rat IgG1 (Abcam) (both at 1 µg/20 g b.w. in 200 µl sterile saline) followed by intratracheal instillation of 5*10$^6$ CFU live E. coli (American Type Culture Collection, ATCC 25922) in 50 µl saline. In separate groups of mice, first pneumonia was induced by intratracheal instillation of live *E. coli*, followed 6 h later (at the peak of inflammation) by intraperitoneal injection of carrier–free recombinant mouse IFN-β1 (Biolegend) (50 ng/20 g b.w. in 200 µl sterile saline) or vehicle. At 6, 24 or 48 h post-E. coli, the lungs were lavaged (4 times with 1 ml heparinized saline) or processed without lavage. Aliquots of homogenized lung were cultured to evaluate *E. coli* colony numbers. Bronchoalveolar lavage (BAL) fluid protein, and total and differential leukocyte counts were determined using standard techniques. Apoptosis in neutrophils (identified as Ly6G-positive cells) was assessed using flow cytometry with FITC-conjugated annexin-V (BD Biosciences). The percentage of macrophages containing apoptotic bodies was assessed following cell staining with hematoxylin and eosin. Lungs removed without lavage were used to determine dry-to-wet weight ratio and tissue myeloperoxidase activity, an index of PMN infiltration. Myeloperoxidase activity was measured using o-dianisidine as a substrate and human MPO (Sigma) as a standard.

**Human neutrophil culture.** Neutrophils were isolated from peripheral blood by centrifugation through a Ficoll-Hypaque gradient, sedimentation through dextran (3% wt/vol), and hypotonic lysis of erythrocytes. Neutrophils (5 × 10$^6$ cells/ml, purity > 95%, viability > 98%, apoptotic < 2%) were cultured in RPMI 1640 medium supplemented with 10% autologous serum with human recombinant IFN-β (25–150 ng/ml, PeproTech) and then challenged with CpG DNA (1.6 µg/ml, E. coli strain B, Sigma), LPS (10 µg/ml, E. coli O111:B4, Sigma), serum amyloid A (10 µg/ml, PeproTech) with or without fludarabine (25 µM) or WP1066 (5 µM). In some experiments, neutrophils were first challenged with CpG DNA and treated with IFN-β at 1–4 h later. To study phagocytosis-induced apoptosis, neutrophils were cultured with live E. coli (American Type Culture Collection, ATCC 25922) at a ratio of 1:7 with or without IFN-β (50 ng/ml). At the designated time points, cells were processed as described below.

**RNA-Seq.** Total RNA from sorted cells was extracted using Aurum Total RNA kit (Bio-Rad Laboratories, Inc.). RNA integrity score was determined by Agilent 2100

Bioanalyzer using the Agilent RNA 6000 Pico kit (Agilent Technologies). Samples were prepared for Illumina sequencing using NEB's Ultra Directional RNA Library Prep Kit for Illumina (NEB#7420). Libraries were sequenced with a 50 bp SR run on Illumina HiSeq 2500 using a V3 flow cell. Sequenced reads were compared to available murine Ensembl 70 genes using mouse genome build (GRCm38), and expression was compared between macrophage subtypes using two separate analysis pipelines: RSEM/EdgeR and topaht2/cuffdiff. Depending on the pipeline, between ~1500–2500 genes were found to be differentially expressed (FDR < 0.05), with a wide overlap in results between the two pipelines. Significance values presented were from the topaht2/cuffdiff analysis. Gene ontology enrichment analysis was performed with the DAVID Bioinformatics Resources 6.7 software.

**Flow cytometry.** Macrophages were recovered from peritoneal exudates of *Ifnb*$^{+/+}$ mice 48–66 h PPI and immunostained immediately or incubated (1 × 10$^6$ cells in 1 mL of culture media) with IFN-β (20 ng/ml) for 48 h. Then, the cells were immunostained with PE anti-mouse F4/80 and PerCP anti-mouse/human CD11b Abs (0.2 µg per million cells each in 100 µl, BioLegend). For detection of intracellular IFN-β protein, peritoneal cells were fixed (1% PFA/PBS /15'/RT), permeabilized (0.05% Tween-20 in 2.5%BSA, 30 min on ice) and immunostained with rat anti-IFNβ (1ug per million cells in 100 µl, Abcam ab24324) and anti-rat Alx488 as a secondary antibody (0.5 µg per million cells in 100 µl, Invitrogen A21208). Cell populations were evaluated by flow cytometry using FACSCanto II (BD) and analyzed by the FlowJo software (Treestar). In some experiments, the phagocyte-specific dye PKH26-PCL red was injected I.P. to mice undergoing peritonitis for 62 h and 4 h later the peritoneal cells were recovered and immunostained as above and with Alexa Fluor 647 anti-mouse CD206 (MMR) (141714), PerCP anti-mouse CD45 (103130) or Alexa Fluor 647 anti-mouse CD36 (102610) Abs (0.2 µg per million cells each in 100 µl, BioLegend) or APC anti-mouse MERTK (10 µl per million cells, R&D Systems FAB5912A). In some experiments, peritoneal cells from unchallenged or zymosan A challenged (48 h), *Ifnb*$^{+/+}$ or *Ifnb*$^{-/-}$ mice were immune-stained with PE-conjugated rat anti-mouse F4/80 and APC-conjugated anti-mouse Tim4 (1:50, Miltenyi Biotec 130–116–758), and analyzed by flow cytometry. Flow cytometry gating strategies are depicted in Supplementary Fig. 8[57].

Surface expression of IFNα/βR1 on human freshly isolated PMN or on neutrophils challenged with CpG DNA (1.6 µg/ml) for 1 or 2 h was assessed using R-phycoerythrin-conjugated mouse anti-human IFNαR1 monoclonal Ab (R&D Systems) and an isotype-matched irrelevant Ab with a FACSCalibur flow cytometer and CellQuestPro software (BD Biosciences). Uptake of FITC-labeled ODN 2395 (Invitrogen) by human neutrophils challenged with IFN-β (12.5–50 ng/ml) or CpG DNA (1.6 µg/ml) for 30 min was assessed by flow cytometry.

**Generation of apoptotic cells.** Jurkat T cells (from Prof. Yablonski, The Technion, Israel) were incubated (1 × 10$^6$ cells/ml of culture media) for 4 h with 1 µM staurosporine (Sigma). In some experiments, cells were then washed with serum-free media and incubated (10 × 10$^6$ cells/ml) for 30 min with 10 mM CypHer5E Mono NHS Ester (GE Healthcare). Then, cells were washed twice with culture medium before incubation with macrophages.

**IFN-β detection.** Peritoneal exudates and bronchoalveolar lavage fluid were collected at the indicated times and the levels of IFN-β in cell-free fluids were determined by VeriKine-HS Mouse Interferon Beta Serum ELISA Kit (Pestka Biomedical Laboratories, Inc). In some experiments, macrophages were recovered 66 h post PPI, separated using PE-conjugated anti-F4/80 Abs (1ul per 10*10$^6$ cells in 100ul, Biolegend 123110) with PE selection magnetic beads (StemCell Technologies) and incubated (1 × 10$^6$ cells in 1 ml of culture media) with TGF-β (5 ng/ml), poly (I:C) (4 µg/ml) or apoptotic cells (1:5 ratio) for 24 h. Then, IFN-β content in conditioned culture media was determined by ELISA.

**Apoptosis assays.** Peritoneal PMN were recovered from *Ifnb*$^{+/+}$ or *Ifnb*$^{-/-}$ mice at 24 h PPI, separated using PE-conjugated Gr1 antibodies with PE selection magnetic beads (StemCell Technologies) and incubated (1*10$^6$ cells in 1 ml of culture media) with or without IFN-β (20 ng/ml) and/or Q-VAD (10 µM) for 24 h. Bronchoalveolar lavage fluid cells were resuspended in 100 µl PBS. 2*10$^5$ cells were stained with Annexin-V-FITC and PI MEBCYTO Apoptosis Kit (MBL Laboratories). Apoptosis was evaluated by flow cytometry using FACSCanto II (BD Biosciences) and analyzed by FlowJo software (Treestar). Alternatively, PMN lysates were prepared after 6 h of culture and immunoblotted for active (cleaved) caspase-3 using specific antibodies (Cell Signaling Technology).

Apoptosis in human PMN was assessed by flow cytometry with FITC-conjugated annexin-V (BD Biosciences) in combination with propidium iodide (Molecular Probes). For nuclear DNA analysis, neutrophils were suspended in 0.2 ml 0.1% sodium citrate solution containing 50 ug/ml propidium iodide and 0.1% Triton X-100 immediately before assay. Mitochondrial transmembrane potential was monitored following neutrophil staining with the lipophilic fluorochrome chloromethyl-X-rosamine (CMXRos, 20 nM, Millipore-Sigma) and the fluorescence was analyzed in a FACSCalibur flow cytometer.

**Ex vivo stimulation for cytokine secretion**. Peritoneal macrophages were recovered from $Ifnb^{+/+}$ or $Ifnb^{-/-}$ mice 48–66 h PPI, separated as above and incubated ($1*10^6$ cells in 1 mL of culture media) overnight with LPS (1 μg/ml). Next, culture supernatants were collected and IL-10, IL-6, IL-12 and CCL3 levels were determined by standard ELISA (Biolegend). In some experiments, macrophages were recovered from peritoneal exudates of $Ifnb^{+/+}$ mice 48–66 h PPI and incubated with IFN-β or IFN-α (20 ng/ml each) for 48 h. Then, culture supernatants were collected and IL-10 and IL-12 levels were determined by standard ELISA. In some experiments, $Ifnb^{+/+}$ or $Ifnb^{-/-}$ mice were injected with vehicle or recombinant mouse IFN-β1 (25 ng/mouse) 24 h prior to exudate recovery and peritoneal macrophages were recovered at 48–66 h PPI.

**Phagocytosis assays in vivo**. The phagocyte-specific dyes PKH2-PCL green or PKH26-PCL red were injected I.P. to mice undergoing peritonitis for 20, 44 and 68 h. In some experiments, apoptotic Jurkat cells, latex beads, IgG-opsonised latex beads ($3*10^6$ particles each) or vehicle were injected together with PKH2-PCL green 62 h after peritonitis initiation. After 4 h, the peritoneal cells were recovered and immuno-stained for F4/80 and CD11b. Alternatively, PKH2-PCL green was injected I.P. to mice undergoing peritonitis for 44 h and at 58 h PKH26-PCL red was injected. At 62 or 66 h post peritonitis initiation, the peritoneal cells were recovered and immuno-stained as just described.

**Immunofluorescence microscopy and image analysis**. For in vivo engulfment evaluation, peritoneal macrophages were recovered from $Ifnb^{+/+}$ or $Ifnb^{-/-}$ mice 48 h PPI, separated as above and fixed with 4% paraformaldehyde + 5% sucrose to 8 well chamber glass slides. Fixed cells were stained overnight at 4 °C with CF 488 phalloidin (1:40; biotium 00042 or 1:1,000; Chem Cruz) followed by Hoechst (1:500; Molecular Probes) and the slides were mounted with SlowFade™ Gold Antifade Mountant (Molecular Probes). The slides were imaged using a Nikon A1-R confocal laser scanning microscope and engulfment was calculated using Nikon NIS-Elements microscope imaging software. For ex vivo engulfment evaluation, macrophages were recovered from $Ifnb^{+/+}$ or $Ifnb^{-/-}$ mice 48 h PPI, separated and incubated ($10*10^6$ Jurkat cells in 1 ml of culture media) with 10 μM CypHer5E Mono NHS Ester-labeled apoptotic Jurkat cells (1:3 ratio, GE Healthcare) and vehicle or IFN-β (25 ng/ml). After 4 h, unbound cells were washed and the cells were stained as above. In some experiments, $Ifnb^{+/+}$ or $Ifnb^{-/-}$ mice were injected with vehicle or recombinant mouse IFN-β1 (25 ng/mouse) 24 h prior to exudate recovery and peritoneal macrophages were recovered at 48–66 h PPI and analyzed as above.

**Organelle fractionation**. RAW264.7 macrophages were incubated with apoptotic Jurkat cells (1 to 3 ratio) for 24 h, washed with PBS and resuspended in 1 ml of ice cold HB buffer supplemented with protease inhibitor cocktail (Roche). The cells were mechanically disrupted in a glass homogenizer and nuclear and cell debris were removed by centrifugation at $110 \times g$ for 5 min at 4 °C. Obtained PNS was carefully applied on top of 4 ml of 10 to 35% OptiPrep (Sigma) gradient and centrifuged for 2 h using SW55 Ti swing Rotor (Beckman Coulter, USA) at $100,000 \times g$ at 4 °C. After centrifugation, 1 ml from the top of the tube was removed and 300 μl fractions were collected thereafter. The collected fractions were mixed with sample buffer, boiled and equal volumes of samples were analyzed using Western blotting.

**Western blot analysis**. Protein extracts of sorted populations (<98% purity) of PKH2-PCL$^{hi}$ and PKH2-PCL$^{lo}$ macrophages were resolved by SDS-PAGE, transferred to PVDF membranes (Bio-Rad Laboratories, Inc.), blocked with 5% skimmed milk powder and immunoblotted with either rabbit anti-human IFN-β Ab (1ug/ml, Abcam ab24324), rabbit anti-mouse ISG-15 (1:200, Santa-Cruz Biotechnology sc-50367) or goat anti-human GAPDH (1:200, Santa-Cruz Biotechnology sc-20357). Lysates from $10^7$ human neutrophils were probed with antibodies to Mcl-1 (1:1,000, Proteintech, cat. No. 16225–1-AP), or YY1 (1:1,000, Santa Cruz Biotechnology, clone H-10, cat. No. sc-7341). For STAT analysis, nuclear and cytosolic fractions from $10^7$ human neutrophils were prepared with a NE-PER Nuclear and Cytosolic Extraction kit (Pierce) and probed with antibodies to phospho-STAT-1 (1:1,000, Cell Signaling, clone 58D6, cat. No. 9167), phospho-STAT3 (1:1,000, Cell Signaling, clone D3A7, cat. No. 9145), YY1 (Santa Cruz Biotechnology) or β-actin (1:1,000, Millipore-Sigma, clone AC-15, cat No. A1978). Alternatively, macrophage lysates, from mice treated with IFN-β (20 ng/mouse, i.p.) 24 h PPI, were immunoblotted for 12/15-LO (1:1,500, Cayman 160704), arginase 1 (1:20,000, Abcam ab60176), ISG15 and GAPDH. Then, the membranes were washed and incubated with the appropriate HRP-conjugated secondary antibodies and developed using WesternBright™ ECL (Advansta Inc) or Clarity Max™ ECL (BioRad). Band density was quantified with National Institutes of Health (NIH) ImageJ software (http://rsb.info.nih.gov/ij/) and was expressed as a ratio of unstimulated cells after correction for loading discrepancies. All unprocessed blots are provided in the Source Data File.

**Statistics**. Statistical significance of differences between indicated samples was determined by unpaired Student's $t$ test or one-way ANOVA followed by Dunn's multiple contrast hypothesis test or Tukey's HSD, using the SPSS software versions 21/24 (IBM) as indicated. Correlations were assessed by the Spearman rank correlation coefficient. $P$ values were indicated as $*p < 0.05$, $**p < 0.01$ and $***p < 0.005$.

**Reporting Summary**. Further information on research design is available in the Nature Research Reporting Summary linked to this article.

## Data availability

RNA sequence data have been deposited in BioProject under the primary accession code PRJNA450293. All other data are available in the article (and its Supplementary Information files), The source data underlying Figs. 1d–h, 2a–k, 3a–i, 4a–m, 5a–f, 6b, c, 6f–h, 7a–i, 8a–i, 9a, b, 9d, e, 9g, 9j, 9l and Supplementary Figs. 1b–f, 2a–g, 3a, b, 4a–c, 5a–r, 6b, c, 7a–g are provided as a Source Data file.

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

## Acknowledgements

*Ifnb*$^{-/-}$ mice were kindly provided by Prof. Shohreh Issazadeh-Navikas, University of Copenhagen, Denmark. This study was supported by grants from the Israel Science Foundation (Grant No. 678/13), the Rosetrees Trust and the Wolfson Family Charitable Trust (to A.A.) and the Canadian Institutes of Health Research (MOP-97742 and MOP-102619) (to J.G.F). S.K.S. is the recipient of the PBC postdoctoral fellowship from the Israeli Council of Higher Education (MALAG), and S.S. is a recipient of a presidential scholarship from the University of Haifa.

## Author contributions

S.K.S, D.E.K, S.S, S.B, M.S, J.S, N.P, S.A, A.O. and S.S-Z performed experiments and analyzed the results. D.E.K and S.S-Z assisted in planning the experiments and writing the manuscript. S.K.S. Y.F, N.S and A.A. performed the genetic and bioinformatics analyses, and deposited the data online. D.B, assisted in image acquisition and analysis. J.G.F and A.A planned the experiments, analyzed the data and wrote the manuscript.

## Additional information

**Competing interests:** The authors declare no competing interests.

