## [Peer Review File · Nature Communications]

Reviewers' comments:

Reviewer #1 (Remarks to the Author):

In this manuscript, Satyanarayanan et al. presented important and novel information of the role of IFN- β in the resolution of acute inflammation by using two different models of acute inflammation: pneumonia induced by *E. coli* and peritonitis by zymosan. In all, the results demonstrate a key role of IFN- β to promote apoptosis and efferocytosis. The study is well conceived, planned, executed and the data are of novelty and interest. Nevertheless, possible misinterpretation and misunderstanding of the data caused by some mistakes and inconsistencies make it hard to understand for readers. Also, the data did not support all the conclusions. Considering the hard work made on planning and performing the experiments, it is strongly recommended to the authors to review the whole manuscript and have it revised.

1. The title of this manuscript is not supported by the experiments presented in the Fig. 3. To consolidate the conclusion that IFN β facilitates the resolution of bacterial inflammation, it is essential to administer exogenous IFN- β to mice after bacterial infection. Indeed, the authors suggest a therapeutic potential for IFN- β in chronic and non-resolving inflammation in the last paragraph of the discussion. However, it is recommended to test this hypothesis in the resolving models applied in this work (bacterial pneumonia or peritonitis) by administering IFN- β at the peak of inflammation and harvesting the samples in a time point that inflammation is still high and unresolved (e.g. 24h). Therefore, the hypothesis of whether IFN- β could anticipate the resolution of acute inflammation must be tested in order to support the general message of this paper.

2. The authors argue that there are two peaks of IFN- β production during peritonitis. However, figure 1g shows statistical difference only between 0h and 4h. Please, confirm if there is a significant difference between times 0 and 72h in terms of IFN- β secretion. Moreover, the levels of IFN- β detected in the BAL fluid (fig 2d) is too low and are below the assay range of the ELISA kit used by the authors (VeriKine-HS Mouse Interferon Alpha- the range is 0.94-60pg/ml).

3- Supplementary fig 1 - Here, the authors should clarify which Figure represents the 4h and 48h time-points indicated in the legend. Additionally, it seems that the 50kDa is expressed by resolving macrophages (probably collected 48h PPI). But what about the exudates, what time they are from? 4h? The authors argue that the 25 kDa was barely detected in macrophages, but it is unclear what time-point was analyzed to draw this conclusion. Taking these findings in consideration - of two different fragments – it is reasonable to hypothesize that the 25kDa fragment is produced during the acute phase of inflammation (~4h) while the 50kDa fragment prevails in the resolution phase. Would this different isoforms have different biological functions in terms of activation of inflammation and resolution? A kinetic of the expression of IFN β in different time-points after zymosan-induced peritonitis in cell extracts and cell-free exudates would clarify the occurrence of these two species of IFN- β . Indeed, the authors must show the data not presented in page 8, lines 2-6.

Furthermore, the authors should discuss the novelty of finding a 50kDa IFN- β species and macrophages as the main producers of IFN- β . Due to the originality of this data, it is recommended the use of a positive control in order to confirm that the presented band surely represents IFN- β . It is also important to use flow cytometry to confirm that IFN- β is from macrophages as stated in the page 9 line 5, since the WB not shown came from a mixed cell population.

4. There is a divergence between the legend of figure 1h and the text (page 8 lines 10-12) in the second section of the results. It is unclear if resolving macrophages were incubated with the stimuli alone or in the presence of apoptotic cells.

5- Figure 3- what is the impact of IFN- β neutralization on bacterial clearance? The authors must present this important data. Also, as suggested above, the administration of exogenous IFN- β after bacterial infection has any impact on bacterial clearance and others parameters evaluated in Fig 3, such as increased apoptosis and efferocytosis? To test this, is recommended to inject

exogenous IFN- β at the peak of bacterial inflammation (6h) and recover the samples at the 24h time-point. To these experiments, I advise to first check the half-life of IFN- β to verify if only one shot will be enough.

6. The authors demonstrate an important function of IFN- β in efferocytosis, which is depicted in figure 6. The authors use IFN knockout mice as a strategy to prove this link. However, it would be important to include the pharmacological treatment of WT and knockout peritoneal macrophages with exogenous IFN- β as a proof of concept. Indeed, in figure 6c it looks like the sum of the frequency of macrophages with 0, 1-3 or >3 neutrophils is higher than 100%. Finally, it is unclear from the ex vivo experiments if IFN has any effect on the rate of efferocytosis (% of efferocytosing macrophages), besides the observed effect in the uptake (neutrophils/macrophages). I would suggest the authors include a graph with the rate of efferocytosis ex vivo.

7. The authors raise the valuable question of whether satiation of macrophages is reversible. However, the results are insufficient to state that satiation is irreversible. Perhaps checking increased, sustained and reduced phagocytosis in IFN- β knockout mice would help the authors to sustain this hypothesis. Indeed, the authors did not mention if diminished efferocytosis would also be irreversible.

8. The link between IFN- β and the resolution of inflammation has been previously suggested by the group of professor Lim and deserves to be mentioned (eg. doi: 10.4049/jimmunol.1301504). Also, it would be important to discuss previous contradictory finding of IFN- β role in chronic inflammation (Type I interferons inhibit the resolution of chronic inflammation. Scheel-Toellner D, Akbar AN, Pilling D, Orteu CH, Buckley CD, Wang K, Webb PR, Lord JM, Salmon M. *Symp Soc Exp Biol.* 2000;52:277-88).

Minor points:

1. Please specify the materials and methods for isolation of cytosolic and nuclear fractions for WB analysis.

2. Please, indicate the software used for the statistical analysis.

3. Pag 3 line 6- I suggest the authors replace the word transform for skew or reprogramming

4. Pag 3 second paragraph - Neutrophil apoptosis has emergedand distinguishes inflammation from its resolution. This is a confusing phrase, please rewrite for clarity.

5. Pag 3 second paragraph - Regarding AnxA1, the given reference (number 16) is not appropriated, since the effect verified was in the pro-apoptotic protein BAD. One appropriate reference that has verified the effect of AnxA1 or its peptide Ac2-26 on MCL-1 accumulation should be given here.

6. Pag 4 - first paragraph -The background of IFN in resolution of inflammation needs to be rewritten for clarity.

7. Pag 7 – line 18 – I would suggest to include: In addition to the predicted ISG-15 MW from 15kDA, it was also detected...

8. Legend of Figure 4- Only YY1 was mentioned as a loading control. However, in the densitometry presented in the fig 4m, the y axis says b-actin. Please, adjust the Figure legend to include b-actin. Please, also explain why YY1 was used for normalization.

9. Pag 13- line 7- the authors assumed that IFN- β overrode the pro-survival effect of CPG DNA in a concentration-dependent manner. However, it does not seem a concentration-dependent effect. There is no statistical indication between the two different concentrations of IFN- β used. The authors should indicate the statistical differences in the graphs of the figures 4g-d.

10. Page 13-line 13. Please change the expression 'a key regulator of apoptosis' for 'a key regulator of neutrophil survival or life-span'.

11. page 17 – line 13-15. Is it a data not shown? Please, clarify.

12- Supplementary Fig 2 legend- The title is not appropriate since the effect verified is indirect. 'IFN- β deficiency affects recruitment...' would fit better with the presented data.

Reviewer #2 (Remarks to the Author):

This article describes a previously poorly-understood role for IFN β in regulating the inflammatory response and phagocytic clearance of apoptotic cells by macrophages during tissue inflammation. Overall, the experiments are well-executed and described appropriately. Moreover, the potential role of IFN β production by tissue macrophages and how this impacts the clearance of apoptotic cells is novel and would be of use to a wide range of immunology researchers. However, the findings presented here seem somewhat preliminary in the sense that it appears that IFN β plays numerous roles in controlling tissue inflammation, and the work shown touches on multiple, distinct functions of IFN β , including PMN apoptosis and macrophage efferocytosis, but without explaining mechanistically how IFN β is functioning to control inflammation *in vivo*. Thus, while this work clearly shows IFN β is a key player in inflammation, the cells that produce IFN β , respond to IFN β , or how IFN β impacts efferocytosis *in vivo* are unclear.

MAJOR:

1. Fig 1A using PKH-PCL dye to identify "satiated" macrophages. However, the authors do not show that these macrophages are in fact "satiated" or otherwise ineffective at uptake of apoptotic cells *in vivo*. Moreover, it seems that these macrophages are not resident peritoneal macrophages but are in fact recruited macrophages, as the authors allude to on p7. However, there is no attempt to separate out the resident (Gata6+, yolk-sac derived macrophages) from recruited macrophages in these experiments. It could be that the PKH+ cells in Fig1A are in YS macrophages while the PKH-/low macrophages are recruited inflammatory macrophages that are poorly phagocytic. The gene expression profiling supports this. So I object to the use of the term "satiated" in the absence of data showing that the macrophages labeled as such in Fig1A are in fact less phagocytic.
2. A major claim of this work is that resident macrophages produce IFN β and that this both enhances PMN apoptosis and macrophage engulfment of these dying cells. However, they do not show that these macrophages release IFN β . They do show that they express it (by mRNA and western blotting in Fig1), but I did not see where they demonstrated IFN β was released by macrophages. They cite data not shown to this effect (p8) – need to see these results to show that macrophages do release IFN β .
3. Are there more/less macrophages in peritoneum of IFN β KO mice?
4. Authors use the term "satiated" macrophage but never show that these macrophages are in fact unable to engulf apoptotic cells.

MINOR:

1. In Fig6, the authors note that there is a higher percentage of macrophages in IFN β KO mice that engulf zero PMN compared to WT mice. For this to be a meaningful readout for cell clearance, it is important to show whether IFN β deletion impacts the numbers and fractions of macrophage populations in the peritoneum in the absence and presence of inflammation.
2. P15, the authors say that IFN β macrophages "appeared more spread and activated" but do not cite shown or unshown data for this. Need to see data supporting this.

Reviewer 1:

- 1. The title of this manuscript is not supported by the experiments presented in the Fig. 3. To consolidate the conclusion that IFN β facilitates the resolution of bacterial inflammation, it is essential to administer exogenous IFN- β to mice after bacterial infection. Indeed, the authors suggest a therapeutic potential for IFN- β in chronic and non-resolving inflammation in the last paragraph of the discussion. However, it is recommended to test this hypothesis in the resolving models applied in this work (bacterial pneumonia or peritonitis) by administering IFN- β at the peak of inflammation and harvesting the samples in a time point that inflammation is still high and unresolved (e.g. 24h). Therefore, the hypothesis of whether IFN- β could anticipate the resolution of acute inflammation must be tested in order to support the general message of this paper.**

We thank the reviewer for raising this important question. We have performed the suggested experiments and presented the results in new Fig. 8 and new Fig. S7. Overall, our results show that treatment of mice with exogenous IFN β at the peak of inflammation enhanced *E. coli* clearance as well as the resolution of inflammation by reducing PMN count in the lavage fluid, promoting monocyte recruitment, PMN apoptosis and efferocytosis (Fig. 8). Furthermore, we also found that exogenous IFN β restored impaired efferocytosis and macrophage reprogramming in IFN β -deficient mice and enhanced these events in WT mice in the peritonitis model (Fig. S7). The Methods (page 34, 5th line; page 38, last line; page 39, 3rd line; page 40, 5th line), Results (Page 23, first line) and figure legends (page 22) have been expanded to describe these results.

- 2. The authors argue that there are two peaks of IFN-beta production during peritonitis. However, figure 1g shows statistical difference only between 0h and 4h. Please, confirm if there is a significant difference between times 0 and 72h in terms of IFN-beta secretion. Moreover, the levels of IFN-b detected in the BAL fluid (fig 2d) is too low and are below the assay range of the ELISA kit used by the authors (VeriKine-HS Mouse Interferon Alpha- the range is 0.94-60pg/ml).**

We thank the reviewer for noting these mistakes. We performed statistical comparison to confirm differences between times 0 and 72 h (please see revised Fig. 1g). We apologize for the misidentified units for IFN β , this has been corrected in the revised Fig. 2d.

- 3. Supplementary fig 1 - Here, the authors should clarify which Figure represents the 4h and 48h time-points indicated in the legend. Additionally, it seems that the 50kDa is expressed by resolving macrophages (probably collected 48h PPI). But what about the exudates, what time they are from? 4h? The authors argue that the 25 kDa was barely detected in macrophages, but it is unclear what time-point was analyzed to draw this conclusion. Taking these findings in consideration - of two different fragments - it is reasonable to hypothesize that the 25kDa fragment is produced during the acute phase of inflammation (~4h) while the 50kDa fragment prevails in the resolution phase. Would these different isoforms have different biological functions in terms of activation of inflammation and resolution? A kinetic of the expression of IFN β in different time-points after zymosan-induced peritonitis in cell extracts and cell-free exudates would clarify the occurrence of these two species of IFN-b. Indeed, the authors must show the data not presented in page 8, lines 2-6.**

Furthermore, the authors should discuss the novelty of finding a 50kDa IFN-b species and macrophages as the main producers of IFN-b. Due to the originality of this data, it is recommended the use of a positive control in order to confirm that the presented band surely represents IFN-b. It is also important to use flow cytometry to confirm that IFN-b is from macrophages as

stated in the page 9 line 5, since the WB not shown came from a mixed cell population.

We thank the reviewer for pointing out this important issue. We performed additional experiments and incorporated the results in the revised Fig. S2. Our new immunoblots show that the major bands found in both resident and resolution phase macrophages have the molecular weight of 50-66 kDa and are absent in IFN β ^{-/-} mice. We obtained similar results by intracellular staining of peritoneal macrophages. Peritoneal fluids on the other hand contained primarily isoforms of 25-35 kDa of IFN β both during the onset and resolution of inflammation. Depleting resident peritoneal macrophages with clodronate-containing liposomes resulted in significant reductions in peritoneal IFN β (25-35 kDa) level. However, administering clodronate-containing liposomes during the resolution phase did not result in a significant reduction in peritoneal IFN β , probably because satiated macrophages that do not acquire clodronate-liposomes were not efficiently eliminated. Finally, using organelle fractionation of RAW264.7 macrophages treated with apoptotic cells, we detected significant differences in the distribution of the various IFN β isoforms. Thus, the 25, 35, 50 and 66 kDa isoforms were present in the ER, whereas only the 50 and 66 kDa isoforms were detectable in secretory vesicles. We interpret the findings that the 25-35 kDa IFN β undergoes posttranslational modifications in the ER and the higher molecular weight products are not secreted until the modifications are removed. Consistently, the 25-35 kDa form is the most abundant IFN β isoform present in body fluids, whereas the 50-66 kDa forms are readily detectable in cell lysates. These results are described on page 8, 2nd line, and briefly discussed on page 32, 2nd line. We are currently investigating whether conjugation of ISG15 to the 25-35 kDa IFN β (ISGylation) is responsible for the generation of 50-66 kDa IFN β and prevention of IFN β secretion. However, we feel that this was beyond the scope of our current manuscript.

4. There is a divergence between the legend of figure 1h and the text (page 8 lines 10-12) in the second section of the results. It is unclear if resolving macrophages were incubated with the stimuli alone or in the presence of apoptotic cells.

We regret for the confusion. We have corrected the text to indicate that macrophages were incubated with only one stimulus at a time (i.e. apoptotic cells, TGF β , or poly I:C).

- 4. Figure 3- what is the impact of IFN-b neutralization on bacterial clearance? The authors must present this important data. Also, as suggested above, the administration of exogenous IFN-b after bacterial infection has any impact on bacterial clearance and others parameters evaluated in Fig 3, such as increased apoptosis and efferocytosis? To test this, is recommended to inject exogenous IFN-b at the peak of bacterial inflammation (6h) and recover the**

samples at the 24h time-point. To these experiments, I advise to first check the half-life of IFN- β to verify if only one shot will be enough.

We thank the reviewer for raising this important issue. We have performed the requested experiments and analyzed the samples at 24h and 48h post-*E. coli*. The results are presented in new Fig. 3a and Fig. 8a. The results show that neutralization of IFN β delayed *E. coli* clearance at 24 h, whereas treatment with exogenous IFN β accelerated bacterial clearance. These experiments are described in the results section (page 11, 2nd line), as well as in the sections indicated in comment 1. We have extended the Discussion to address the implication of these findings.

6. The authors demonstrate an important function of IFN- β in efferocytosis, which is depicted in figure 6. The authors use IFN knockout mice as a strategy to prove this link. However, it would be important to include the pharmacological treatment of WT and knockout peritoneal macrophages with exogenous IFN- β as a proof of concept. Indeed, in figure 6c it looks like the sum of the frequency of macrophages with 0, 1-3 or >3 neutrophils is higher than 100%. Finally, it is unclear from the ex vivo experiments if IFN has any effect on the rate of efferocytosis (% of efferocytosing macrophages), besides the observed effect in the uptake (neutrophils/macrophages). I would suggest the authors include a graph with the rate of efferocytosis ex vivo.

We thank the reviewer for this important comment. We have performed the requested experiments and the results are presented in Fig. 8 and supplementary Fig. 5. The results show that treatment with exogenous IFN β enhanced the uptake of apoptotic PMN by macrophages during the resolution of pneumonia (Fig. 8i), and rescued the defective uptake of apoptotic PMN in IFN $\beta^{-/-}$ mice during the resolution of peritonitis (Fig. S7). We also corrected the numbers in Fig. 6c and included the percentage of efferocytosing macrophages ex vivo in Fig. 6g.

7. The authors raise the valuable question of whether satiation of macrophages is reversible. However, the results are insufficient to state that satiation is irreversible. Perhaps checking increased, sustained and reduced phagocytosis in IFN- β knockout mice would help the authors to sustain this hypothesis. Indeed, the authors did not mention if diminished efferocytosis would also be irreversible.

We thank the reviewer for pointing out this important issue. While we agree that the suggested experiment might lend additional support to our hypothesis, unfortunately we were unable to perform these experiments within the past 6 months allowed for revision by the Editor because of the current shortage of the PKH2-green reagent from the manufacturer, and the large number of IFN $\beta^{-/-}$ mice needed to perform this complex analysis. However, we did assess PKH2 uptake in IFN $\beta^{-/-}$ mice and detected similar percentage of satiated macrophages as in their

WT counterparts. However, we realize that macrophage percentages might be affected by factors other than satiation (e.g., monocyte infiltration, macrophage emigration and apoptosis). To address more directly the reviewer's comment, we co-injected labeled apoptotic cells, latex beads or IgG-opsonized latex beads together with PKH2-red, and analyzed the recovered F4/80+ macrophages for concomitant target uptake and PKH2 engulfment. The new Fig. 9c-e show that only phagocytic macrophages that acquired high levels of PKH2 were able to accumulate latex beads or IgG-latex beads. By contrast, neither PKH2-lo nor PKH2-negative macrophages phagocytosed these particles. Apoptotic cells were also mostly acquired by PKH2-hi macrophages, albeit we detected some uptake by PKH2-medium macrophages. PKH2-negative macrophages did not uptake apoptotic cells either. These results would indicate that macrophages that were satiated at the assay initiation were non-phagocytic at all. It seems that some macrophages that acquired apoptotic neutrophils during the assay might have become satiated, and consequently accumulated less apoptotic cells and PKH2 than PKH2-hi macrophages that remained phagocytic throughout the assay. We believe these findings unequivocally support our initial findings that satiation is irreversible. These experiments are described in the results section (page 24, 12th line).

8. The link between IFN- β and the resolution of inflammation has been previously suggested by the group of professor Lim and deserves to be mentioned (eg. doi: 10.4049/jimmunol.1301504).

We agree and have revised the discussion on page 28 line 4, as suggested.

9. Also, it would be import to discuss previous contradictory finding of IFN- β role in chronic inflammation (Type I interferons inhibit the resolution of chronic inflammation. Scheel-Toellner D, Akbar AN, Pilling D, Orteu CH, Buckley CD, Wang K, Webb PR, Lord JM, Salmon M. *Symp Soc Exp Biol.* 2000;52:277-88).

We thank the reviewer for pointing out the apparently contradictory clinical findings regarding the role of type I IFNs in chronic inflammation. We have briefly discussed these findings on page 31, lines 1-14. In our opinion, there are major differences between the resolution of chronic and acute inflammation is the cell types and signaling pathways targeted by IFN β . In chronic inflammation, IFN β activates STAT1 in T cells to promote survival, while in the acute setting it acts through STAT3 in PMN to promote apoptosis. Four new references have been included to support the revised Discussion.

Minor points:

1. Please specify the materials and methods for isolation of cytosolic and nuclear fractions for WB analysis.

This has been included in the revised version (page 42, 2nd line).

2. Please, indicate the software used for the statistical analysis.

This information was added on page 44, 6th line from bottom.

3. Pag 3 line 6- I suggest the authors replace the word transform for skew or reprogramming

Since the word reprogramming already appears in this sentence we preferred to use **diverts**.

4. Pag 3 second paragraph - Neutrophil apoptosis has emergedand distinguishes inflammation from its resolution. This is a confusing phrase, please rewrite for clarity.

The sentence has been rephrased as "Neutrophil apoptosis is currently perceived as one of the control points that limits neutrophil numbers at sites of inflammation and pushes ongoing inflammation toward its resolution".

5. Pag 3 second paragraph - Regarding AnxA1, the given reference (number 16) is not appropriated, since the effect verified was in the pro-apoptotic protein BAD. One appropriate reference that has verified the effect of AnxA1 or its peptide Ac2-26 on MCL-1 accumulation should be given here.

The reference was changed to Vago, J.P. et al. Annexin A1 modulates natural and glucocorticoid-induced resolution of inflammation by enhancing neutrophil apoptosis. J Leukoc Biol 92, 249-258 (2012).

6. Pag 4 - first paragraph -The background of IFN in resolution of inflammation needs to be re-written for clarity.

The paragraph has been rewritten.

7. Pag 7 – line 18 – I would suggest to include: In addition to the predicted ISG-15 MW from 15kDA, it was also detected...

The sentence has been changed accordingly.

8. Legend of Figure 4- Only YY1 was mentioned as a loading control. However, in the densitometry presented in the fig 4m, the y axis says b-actin. Please, adjust the Figure legend to include b-actin. Please, also explain why YY1 was used for normalization.

We apologize for this oversight. We used the ubiquitous transcription factor YY1 as a loading control. The difference in the molecular mass of YY1 (68KDa) and Mcl-1 (37kDa) permits simultaneous visualization of these proteins after cutting the membranes without stripping.

9. Page 13- line 7- the authors assumed that IFN-b overrode the pro-survival effect of CPG DNA in a concentration-dependent manner. However, it does not seem a concentration-dependent effect. There is no statistical indication between the two different concentrations of IFN-b used. The authors should indicate the statistical differences it in the graphs of the figures 4g-d.

Statistically significant differences are indicated on the revised Fig. 4g-d.

10. Page 13-line 13. Please change the expression ‘a key regulator of apoptosis’ for ‘a key regulator of neutrophil survival or life-span’.

The change was made as requested.

11. page 17 – line 13-15. Is it a data not shown? Please, clarify.

We regret causing confusion. These results were previously presented in Fig. S5, but this was erroneously not indicated in the text. The data are presented in the revised Fig. S7, which also depicts rescue of the IFN β ^{-/-} macrophage phenotype by exogenous IFN β . The text has been modified accordingly (page 23, line 15 in the revised version).

12- Supplementary Fig 2 legend- The title is not appropriate since the effect verified is indirect. ‘IFN-b deficiency affects recruitment...’ would fit better with the presented data.

The title has been changed as requested.

Reviewer 2:

- 1. Fig 1A using PKH-PCL dye to identify “satiated” macrophages. However, the authors do not show that these macrophages are in fact “satiated” or otherwise ineffective at uptake of apoptotic cells in vivo. Moreover, it seems that these macrophages are not resident peritoneal macrophages but are in fact recruited macrophages, as the authors allude to on p7. However, there is no attempt to separate out the resident (Gata6+, yolk-sac derived macrophages) from recruited macrophages in these experiments. It could be that the PKH+ cells in Fig1A are in YS macrophages while the PKH-/low macrophages are recruited inflammatory macrophages that are poorly phagocytic. The gene expression profiling supports this. So I object to the use of the term “satiated” in the absence of data showing that the macrophages labeled as such in Fig1A are in fact less phagocytic.**

We thank the reviewer for this important comment. We performed additional experiments to document that the in vivo uptake of apoptotic cells and other targets, such as latex beads or IgG-opsonized latex beads is indeed down-regulated or completely absent in macrophages that acquire low levels of PKH2 (please see new Fig. 9c-e.) These results show that not only do PKH2^{lo/-} macrophages acquire low levels to none of any targets examined. Furthermore, the levels of acquisition of latex beads or IgG-latex beads correlates nicely with the level of PKH2 uptake. These findings would indicate that PKH2^{lo/-} macrophages are indeed satiated, and the uptake of PKH2 truly reflects their phagocytic capacity. The corresponding text has been expanded on page 23, line 8. Concerning the origin of the satiated macrophages, we have included data with 2 additional markers of resident peritoneal macrophages, Tim4 and GPR37, in addition to GATA6 and TGF β 2 (that were shown in the original submission). However, Tim4 and GPR37 mRNA is

barely detectable or non-detectable, respectively, in both macrophage subsets (please see text on page 6, 3rd line), lending support to our claim that both subsets are monocyte-derived. Moreover, immuno-staining of resident and resolution phase macrophages for F4/80 and Tim4 revealed that in accordance with our RNA-Seq analysis, about 5% of macrophages present during the resolution phase originated from resident macrophages. By contrast, the phagocytic macrophages constitute 60-70% of the resolution phase macrophage population. These results are presented in the revised Fig. S1 and described in the text on page 7, 1st line. We interpret these findings that both phagocytic and satiated macrophages are indeed originating from monocyte-derived phagocytic macrophages rather than resident macrophages.

2. A major claim of this work is that resident macrophages produce IFN β and that this both enhances PMN apoptosis and macrophage engulfment of these dying cells. However, they do not show that these macrophages release IFN β . They do show that they express it (by mRNA and western blotting in Fig1), but I did not see where they demonstrated IFN β was released by macrophages. They cite data not shown to this effect (p8) – need to see these results to show that macrophages do release IFN β .

We thank the reviewer for pointing out this important aspect of our work. To demonstrate macrophage release of IFN β , we have included results previously referred to as data not shown together with new data of intracellular staining of leukocytes recovered at the resolution phase of inflammation for IFN β . We now show (Fig. S2) that macrophages from WT, but not IFN $\beta^{-/-}$ mice contain detectable levels of IFN β , while peritoneal eosinophils and lymphocytes stained negative for IFN β . Furthermore, depletion of macrophages using clodronate-containing liposomes confirmed that these cells were indeed responsible for the production of IFN β during the different phases of peritonitis.

3. Are there more/less macrophages in peritoneum of IFN β KO mice?

We thank the reviewer for this important comment. We compared macrophage counts in WT and IFN $\beta^{-/-}$ mice and detected lower numbers of peritoneal macrophages at 48 hrs PPI in the latter ones. These results are presented as new Fig. 6h and are described in the results section (page 17, 2nd line from bottom).

4. Authors use the term “satiated” macrophage but never show that these macrophages are in fact unable to engulf apoptotic cells.

Please see our response to comment 1.

MINOR:

1. In Fig6, the authors note that there is a higher percentage of macrophages in IFN β KO mice that engulf zero PMN compared to WT mice. For this to be a meaningful readout for cell clearance, it is important to show whether IFN β deletion impacts the numbers and fractions of macrophage populations in the peritoneum in the absence and presence of inflammation.

As indicated in comment 3, IFN β ^{-/-} mice have lower numbers of macrophages in the peritoneum similar to that reported by the Reeves group for IFN α 1^{-/-} mice in pristane-induced inflammation. Therefore, it is unlikely that differences in efferocytosis in vivo can be attributed to increased macrophage numbers that results in reduced average availability of PMN. Our ex vivo results further underscore this point as the ratio of macrophages to apoptotic cells was kept constant in all assays.

2. P15, the authors say that IFN β macrophages “appeared more spread and activated” but do not cite shown or unshown data for this. Need to see data supporting this.

We apologize for this oversight. We added arrowheads to the images in Fig. 6e to indicate the activated and spread macrophages and revised the text accordingly on page 17, 5th line from bottom.

We thank you and the reviewers for the constructive comments that were helpful in revising the manuscript, and, in our opinion, considerably improved it. We trust you will find our revised manuscript in good order and suitable for publication in Nature Communications.

Reviewers' comments:

Reviewer #1 (Remarks to the Author):

The authors have answered satisfactorily to all my questions and performed additional experiments that really improved the quality of the work. In my opinion, the manuscript is now suitable to publication in Nature communications. I just recommend that the authors update the abstract including a phase that describes the new important result after revision, exogenous administration of IFN- β promotes resolution of inflammation and promotes bacterial clearance, which is fundamental to fit with the title.

There are some minor errors in text as follow:

-Supplementary fig 2C- The figure and legend say 16h peritoneal fluids- the manuscript (page 8 line 4) says 4h.

-Legend of The Fig 7g says that experiment was performed in vitro- Macrophages were recovered from peritoneal exudates of IFN β +/+ mice 48-66 h PPI and incubated with IFN- β (20 ng/ml) for 48 h., while in the text page 20 line 20 says in vivo.

Reviewer #2 (Remarks to the Author):

The new information provided by the authors satisfactorily address the concerns that I enumerated from the first submission EXCEPT the following:

1. The authors continue to use the term "satiated" to describe the phagocytic activity of the PKH2LO peritoneal macrophages. While it is clear that these PKH2LO macrophages show reduced/negligible phagocytic activity in their experimental systems, they do not prove that the "satiation" they claim is due to the engulfment of apoptotic cells (i.e. infiltrating neutrophils) during inflammation/resolution. The new data provided in Fig. 9c-e they proves that apoptotic cells can induce satiation (as stated in text on pg. 24, line 20). However, as stated in the legend for Fig. 9, in this experiment the apoptotic cells were "applied together with PKH2 dye" i.p. rather than apoptotic cells being applied first followed by the PKH dye. Thus, the most simple explanation for the fact that the PKH2LO cells do not engulf apoptotic cells is that the cells they have identified as "satiated" are simply a cell population that has weak phagocytic activity (for both apoptotic cells and the PKH dye). Based on all of the data presented in this manuscript, I see no convincing evidence that the PKH2LO macrophages are satiated due to engulfment of PMN during inflammation as they claim, but instead are simply macrophages that have intrinsically reduced phagocytic activity. For these reasons, I feel strongly that the idea the authors put forward that these macrophages are "satiated" as a result of engulfment of PMN during inflammation is simply not supported by the data shown and rather reflects a very loose interpretation of the data shown. If the authors wish to validate this claim, they must show that the PKH2LO / "satiated" macrophages enter this state due to efferocytosis of PMN during inflammation; alternatively, the use of a more objective appellation for this population –one that is consistent with the data presented – is necessary to avoid misrepresenting the data and adding confusion to the literature on this topic.

2. In Major Point 3, I requested that the authors show data for the baseline macrophage populations in WT vs IFN β KO mice. In their rebuttal to this point they indicate that they have added these data in Fig. 6h showing macrophage numbers 48hrs post-inflammation. I think it is important to know if there are differences in the numbers and phenotype of resident macrophages in healthy WT vs IFN β KO mice (i.e. before induction of inflammation).

Reviewer 1:

1. **I just recommend that the authors update the abstract including a phase that describes the new important result after revision, exogenous administration of IFN- β promotes resolution of inflammation and promotes bacterial clearance, which is fundamental to fit with the title.**

Thank you for the suggestion, we have revised the abstract accordingly.

2. **Supplementary fig 2C- The figure and legend say 16h peritoneal fluids- the manuscript (page 8 line 4) says 4h.**

The figure legend was correct, the text has been corrected accordingly.

3. **Legend of The Fig 7g says that experiment was performed in vitro- Macrophages were recovered from peritoneal exudates of IFN β +/+ mice 48-66**

h PPI and incubated with IFN- β (20 ng/ml) for 48 h., while in the text page 20 line 20 says *in vivo*.

The figure legend was correct, the text has been corrected accordingly.

Reviewer 2:

The authors continue to use the term “satiated” to describe the phagocytic activity of the PKH2LO peritoneal macrophages. While it is clear that these PKH2LO macrophages show reduced/negligible phagocytic activity in their experimental systems, they do not prove that the “satiation” they claim is due to the engulfment of apoptotic cells (i.e. infiltrating neutrophils) during inflammation/resolution. The new data provided in Fig. 9c-e they proves that apoptotic cells can induce satiation (as stated in text on pg. 24, line 20). However, as stated in the legend for Fig. 9, in this experiment the apoptotic cells were “applied together with PKH2 dye” i.p. rather than apoptotic cells being applied first followed by the PKH dye. Thus, the most simple explanation for the fact that the PKH2LO cells do not engulf apoptotic cells is that the cells they have identified as “satiated” are simply a cell population that has weak phagocytic activity (for both apoptotic cells and the PKH dye). Based on all of the data presented in this manuscript, I see no convincing evidence that the PKH2LO macrophages are satiated due to engulfment of PMN during inflammation as they claim, but instead are simply macrophages that have intrinsically reduced phagocytic activity. For these reasons, I feel strongly that the idea the authors put forward that these macrophages are “satiated” as a result of engulfment of PMN during inflammation is simply not supported by the data shown and rather reflects a very loose interpretation of the data shown. If the authors wish to validate this claim, they must show that the PKH2LO / “satiated” macrophages enter this state due to efferocytosis of PMN during inflammation; alternatively, the use of a more objective appellation for this population –one that is consistent with the data presented – is necessary to avoid misrepresenting the data and adding confusion to the literature on this topic.

We agree with the reviewer that our data do not demonstrate directly the role of apoptotic neutrophil uptake *in vivo*. However, we used the term "satiated" macrophages to simply describe macrophages that were previously phagocytic and subsequently lost their phagocytic capacity, as have been demonstrated throughout this manuscript and in that by Schif-Zuck et al. (Eur J Immunol 2011, Ref #9). We believe this term does not indicate the trigger(s) for satiation. However, to improve the description of our findings, we have changed "satiated" to "non-phagocytic" macrophages, and "satiation" to "loss of phagocytosis". We have also “toned” down the role of neutrophil uptake by macrophages in their loss of phagocytosis. To further clarify this issue, we have also added a brief section to the discussion on satiation, as we see it, and its potential link to apoptotic cell uptake. These changes are highlighted in yellow throughout the revised manuscript.

2. In Major Point 3, I requested that the authors show data for the baseline macrophage populations in WT vs IFN β KO mice. In their rebuttal to this point they indicate that they have added these data in Fig. 6h showing macrophage numbers 48hrs post-inflammation. I think it is important to know if there are differences in the numbers and phenotype of resident macrophages in healthy WT vs IFN β KO mice (i.e. before induction of inflammation).

We agree and included the requested data as Fig. S1c-f, and described the results on Page 17, last line, and page 20, line 15, in the revised version.

We thank you and the reviewers for the constructive comments that were helpful in revising the manuscript, and, in our opinion, considerably improved it. We trust you will find our revised manuscript in good order and suitable for publication in Nature Communications.

REVIEWERS' COMMENTS:

Reviewer #2 (Remarks to the Author):

The author responses and corresponding edits to the manuscript are satisfactory.